# Spatiotemporal dynamics of ecosystem services in response to climate variability in Maze National Park and its environs, southwestern Ethiopia

**Mestewat Simeon**[1,2]*, **Desalegn Wana**[1], **Zerihun Woldu**[3]

**1** Department of Geography & Environmental Studies, Addis Ababa University, Addis Ababa, Ethiopia,
**2** Department of Geography & Environmental Studies, Hawassa University, Hawassa, Ethiopia,
**3** Department of Plant Biology & Biodiversity Management, Addis Ababa University, Addis Ababa, Ethiopia

* mamemest2015@gmail.com

**Data Availability Statement:** All relevant data are within the manuscript and its Supporting Information files.

**Funding:** This study was financially supported by Addis Ababa University thematic research grant

## Abstract

Climate variability is one of the major factors affecting the supply of ecosystem services and the well-being of people who rely on them. Despite the substantial effects of climate variability on ecosystem goods and services, empirical researches on these effects are generally lacking. Thus, this study examines the spatiotemporal impacts of climate variability on selected ecosystem services in Maze National Park and its surroundings, in southwestern Ethiopia. We conducted climate trend and variability analysis by using the Mann-Kendall (MK) trend test, Sen's slope estimator, and innovative trend analysis (ITA). Relationships among ecosystem services and climate variables were evaluated using Pearson's correlation coefficient (r), while partial correlation was used to evaluate the relationship among key ecosystem services and potential evapotranspiration (PET). The MK tests show a decreasing trend for both mean annual and main rainy season rainfall, with Sen's slope (β) = -0.721 and β = -0.1.23, respectively. Whereas, the ITA method depicted a significant increase in the second rainy season rainfall (Slope(s) = 1.487), and the mean annual (s = 0.042), maximum (s = 0.024), and minimum (s = 0.060) temperature. Spatial correlations revealed significant positive relationships between ecosystem services and the mean annual rainfall and Normalized Difference Vegetation Index (NDVI), while negative correlations with the mean annual temperature. Additionally, temporal correlations highlighted positive relationships among key ecosystem services and the main rainy season rainfall. The maximum and minimum temperatures and ecosystem services were negatively correlated; whereas, there was strong negative correlations between annual (r = -0.929), main rainy season (r = -0.990), and second rainy season (r = -0.814) PET and food production. Thus, understanding the spatiotemporal variability of climate and the resulting impacts on ecosystem services helps decision-makers design ecosystem conservation and restoration strategies to increase the potential of the ecosystems to adapt to and mitigate the impacts of climate variability.

(TR/22/2021). The funder had no role in study design, data collection and analysis, decisions to publish, interpretation of the data and preparation of the manuscript for publication.

## 1. Introduction

Climate change significantly affects both the quality and quantity of ecosystem services [1] by increasing the frequency and intensity of wildfires, floods, crop failures, and outbreaks of disease and insect damages [2–4]. Ecosystem services are defined as the benefits society obtains from ecosystems, and are categorized as provisioning, regulating, supporting, and cultural services [2]. Nonetheless, climate variability is often negatively affecting these ecosystem goods and services [5], such as food production, drinking water, wood fuel, fodder, climate regulation, and carbon sequestration [1]. This is particularly evident through frequent temperature variability, recurrent droughts, shortened growing seasons [6], and changes in mean temperature and rainfall [7,8] in arid and semi-arid areas [6]. Climate variability is primarily distinguished by trends and fluctuations in the climate's state on spatial and temporal scales at a relatively shorter period of time, while climate change refers to the changes in climate elements over the longer period of time [9]. The extent to which climate change and variability affect ecosystem goods and services are highly localized and varies spatially. Hence, provisioning services such as food production in semi-arid areas are significantly impacted by these factors [5]. Overall, climate change and variability have numerous potentially serious negative impacts on key provisioning and regulatory services obtained from ecosystems [4,10,11].

The changing climate, evidenced by climate variability, affects the provision of ecosystem services and the well-being of people who rely on these services [12]. Ethiopia, like many other African countries, is very susceptible to the negative impacts of climate change and variability due to its low adaptive capacity [13]. In Ethiopia, climate variability due to increasing temperature, and declining and unreliable rainfall have negatively affected ecosystem services such as food production, groundwater availability, and soil organic matter dynamics [14]. In addition, the National Ecosystem Assessment (NEA) report of Ethiopia indicated that the multidimensional impacts of climate change and variability caused the loss of biodiversity and ecosystem services in the country, which are vital for human well-being [13].

With this regard, national parks are crucial for biodiversity conservation and for local residents who depend on natural resources for their survival [15,16]. They provide ambient weather, good pasture, construction wood, charcoal, fuel wood, thatching grass, and cultural services [16,17].The study area, Maze Nation Park (MzNP) and its surroundings, provide the local communities with ecosystem goods such as food, water, pasture, thatching grass, fuelwood, and construction materials and a host of other ecosystem services such as clean water, climate regulations, and cultural and environmental amenities. However, the supply of these vital ecosystem goods and services is dwindling due to climate variability-induced changes, such as recurrent droughts, crop pests, and frequent forest/bush fires [18].

Previous ecosystem goods and services related studies in Ethiopia have primarily focused on evaluating the impacts of land use land cover (LULC) changes on ecosystem services in different parts of the country [19–22]. These studies mainly focused on the relationships between LULC change and ecosystem goods and services, with limited attention given to other significant determining factors. Despite the substantial negative effects of climate variability, including changes in temperature and precipitation, as well as climate variability related disturbances such as flooding, drought, and wildfires on ecosystem goods and services [1,23], empirical research on these effects remains lacking in the country [13]. Few studies for instance, [24] have focused on the impacts of climate variability, particularly the variability of precipitation, on water yield only, without considering other provisioning and regulatory services. In a broader context, previous reviews, for example [10], has addressed the observed and anticipated impacts of climate change on biodiversity and ecosystem services in Africa, but did not include any quantitative analysis of the relationship between these variables. Therefore,

assessing the impacts of climate change and variability on ecosystem services and quantitatively analyzing their associations can provide scientific basis for the protection and restoration of local and regional ecosystems [25]. Our study, therefore, aims at investigating the spatiotemporal impacts of climate variability on the provisioning of key ecosystem services in MzNP and its environs, in southwestern Ethiopia. In this study, we analyzed the temporal and spatial variability of temperature and rainfall and assessed their correlation with selected key ecosystem services, including food production, water supply, raw materials, and climate regulation services. Additionally, we evaluated the relationship between potential evapotranspiration and ecosystem services, as well as between Normalized Difference Vegetation Index (NDVI) and ecosystem services.

## 2. Materials and methods

### 2.1. Description of the study area

The study area, MzNP and its surroundings, is located in Gamo and Gofa zones of the Southern Ethiopia Regional State. The study area is situated about 468 km southwest of Addis Ababa, national capital. It is situated between 06˚11′35″N– 06˚37′49″N Latitudes and 37˚03′42″E– 37˚24′55″E Longitudes (Fig 1). The park and the surrounding seventeen (17) *kebeles* (the smallest administrative unit in Ethiopia) make up the research area. The park, established in 2005 [26], is known for its rich species diversity [27] and for harboring the endangered and endemic subspecies of Swayne's hartebeest [28].

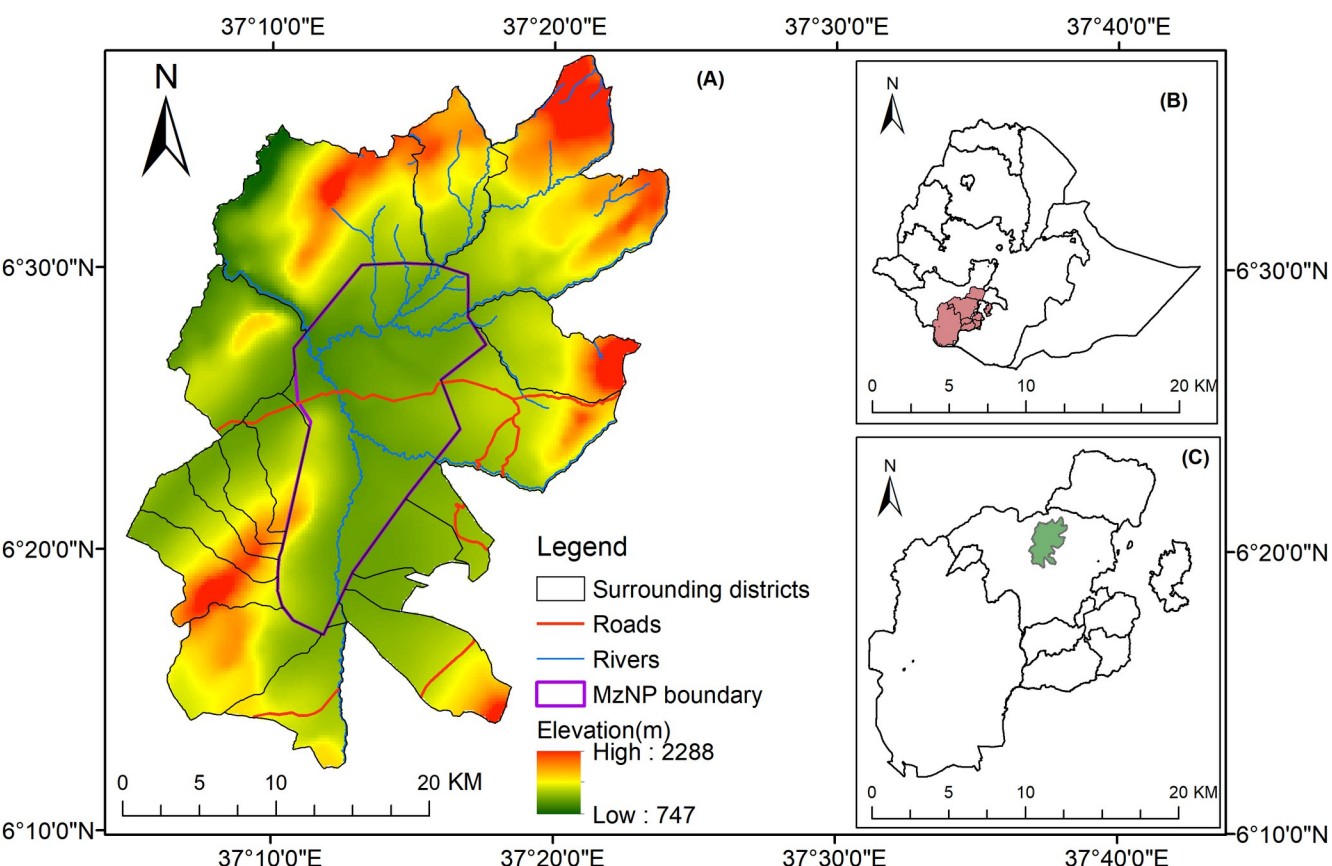

**Fig 1. Location map of the study area.** Note: (A) Maze national park and the surrounding districts (B) location of Southern Ethiopian region in Ethiopia and (C) location of the study area in Southern Ethiopian region.

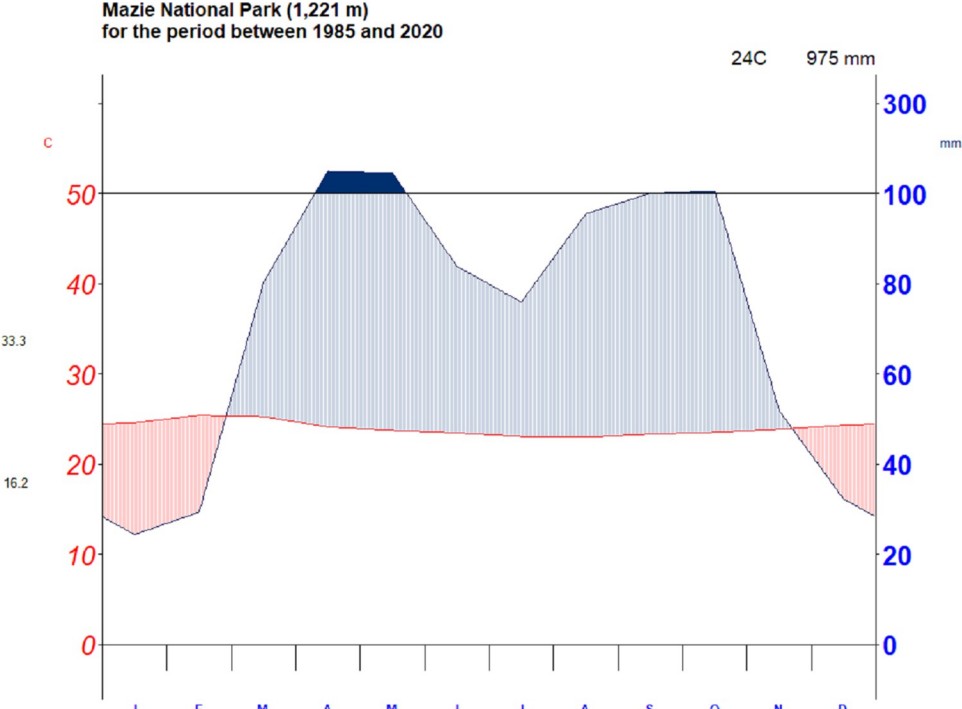

**Fig 2. Walter-Lieth climate diagram of the study area for the period between 1985 and 2020.** Source: National Meteorological Agency (NMA), (2022).

According to the traditional agro-ecological zones of Ethiopia, the study area is categorized under *Kolla (*lowland with an elevation between 500 and 1500 m above sea level) *and Woina-dega (*midland, between 1500–2300 m above sea level) agroecological zones [29]. Its elevation ranges between 747 and 2288 meter above sea level. The landscape of the study area includes a vast plain, some sloppy areas, small hills, and escarpments and is surrounded by highly rugged mountain ranges. Grided data collected from Meteorological Services Agency from 1985–2020 shows that the mean annual temperature of the study area is 23.96˚C. Rainfall exhibited a bimodal type of pattern in the study area, with two distinct rainfall seasons. March to May is the main rainy season (spring), and September to November is the second rainy season (autumn), with the surplus from March to May and in October, respectively (Fig 2). The mean annual rainfall of the study area from 1985 to 2020 was 975 mm (Fig 2). Large parts of the southern regions of Ethiopia experiences a long rainfall season (March to May) with rainfall peaks in April/May and a short rainfall season from October to November [30,31]. The primary means of subsistence of the local communities in the study area is cereal production and livestock rearing. However, tuber crops such as sweet potato (*Ipomoea batatas*), taro (*Colocasia esculenta*), and cassava (*Manihot esculenta*) are cultivated as supplements to cereals [32]. The park and its surrounding environment provide a wide variety of services, including provisioning, regulating, and cultural services for the local communities [33]. Nevertheless, the park faces insurmountable challenges due to LULC changes, and extreme climate events such as recurrent drought and frequent forest fires [34]. These challenges have, therefore, led to declines in the major ecosystem services obtained from the park and its surrounding environs over time [33].

## 2.2. Data sources and methods

**2.2.1. Sources of data.** Temperature and rainfall data were used to analyze climate variability in the study area because these variables have received great attention worldwide to deal with climate variability [35]. The climatic data used in this study was obtained from the Meteorological Services Agency of Ethiopia, southern district office. The study used gridded monthly data of rainfall, maximum temperature, and minimum temperature of seven grid points from varying agroecological settings (*Kola and Woinadega*) covering the period of 36 years (1985–2020). The dataset has a spatial resolution of 4km x 4 km, and the data were reconstructed into series based on records of weather stations and meteorological satellite observations. Grid data was preferred due to the absence of sufficient meteorological stations in the study area due to its remoteness and the critical problem of missing data. The averaged values of these grid points were used to analyze rainfall and temperature variability in the study area.

We conducted Focus group discussions (FGDs) and key informant interviews (KII) with the local community, agricultural extension agents, and park staff to identify four ecosystem goods and services such as food production, water supply, raw material provision, and climate regulatory services (Table 1). These services were considered as major benefits [33] they obtain from the park and were also highly affected by climate variability in the last three decades.

The values of the above selected key ecosystem goods and services were estimated from the LULC maps of 1985, 2005, and 2020 [33] using the benefit transfer method [36,38,39]. In addition to the ESVs for 1985, 2005, and 2020 from the previous study [33], we also estimated the values of the selected services for 1995 and 2015 using a similar method. The benefit transfer method was utilized to estimate the economic value of ecosystem goods and services when specific valuation data were lacking. This approach adapts existing valuation information to suit new policy contexts, especially valuable when limitations such as budget, time, and data availability restrict the collection of primary data[40,41]. This method has extensively been used to value environmental resources in numerous studies[36,38,39]. The LULC classifications were carried out using a Random Forest (RF) supervised classification method in R statistical software (R4.1.3) [42]. Accuracy assessment of the classified images was done via the Semi-Automatic Classification Plugin (SCP) in QGIS using high spatial resolution satellite images and Google Earth images as a reference[33].

We used value coefficients from the ecosystem service valuation database (ESVD) to quantify the values of the identified ecosystem goods and services. The ESVD was updated in 2020 with the support from the UK Department for Environment, Food, and Rural Affairs (Defra) [37]. Estimated values of key ecosystem services are presented in Table 2.

**2.2.2. Measurement of variability.** This study used the coefficient of variation (CV) and rainfall anomaly index (RAI) to analyze temperature and rainfall variability. CV examines the year-to-year variation in the data series. The higher the value of CV, the higher the variability

**Table 1.** *Description of key ecosystem services.*

| Ecosystem Services | Description | Sources |
|---|---|---|
| Food production | Production of crops, nuts, fruits by hunting, gathering, subsistence farming or fishing. | [2,36] |
| Water supply | Provisioning of water by watersheds for drinking, irrigation and other domestic uses | [36,37] |
| Raw materials | The production of lumber, fuel wood, fodder/pasture, charcoal, and thatching grass | [36,37] |
| Climate regulation | Regulation of local temperature and precipitation | [2,36] |

**Table 2. Values of selected key ecosystem services of the study area (US$ million).**

| Year | Ecosystem Services | | | | | |
|------|-----------------|---|-------------|---------------------|--------------------|---------|
| | Food production | | Water supply | Raw material provision | Climate regulation | Total |
| 1985 | 15.85 | | 839.94 | 234.68 | 15.56 | 1106.03 |
| 1995 | 15.30 | | 779.00 | 218.04 | 14.56 | 1026.9 |
| 2005 | 14.60 | | 697.41 | 199.78 | 13.57 | 925.36 |
| 2015 | 16.26 | | 671.81 | 188.57 | 12.67 | 889.31 |
| 2020 | 16.55 | | 659.59 | 188.96 | 12.89 | 877.99 |

of rainfall in the study area, and vice versa. The value of CV can be computed as follows:

$$CV = \frac{\sigma}{\mu} \times 100 \tag{1}$$

Where CV is the coefficient of variation, $\sigma$ is the standard deviation and $\mu$ is the mean. According to [43], the degree of rainfall variability is classified as low variability (CV < 20), moderate variability (20 < CV < 30), and high variability (CV > 30). Therefore, the coefficient of variation was calculated to detect the variation of monthly, seasonal, and annual rainfall and temperature records during the study period.

RAI is used to examine the frequency and intensity of prior dry and wet years [44]. In this study, RAI was employed to identify years of positive and negative anomalies of rainfall fluctuations. Positive rainfall anomalies indicate wet years, while negative rainfall anomalies indicate dry years [44]. RAI was calculated using the following equation:

$$RAI = 3 * \left[ \frac{N - \bar{N}}{\bar{M} - \bar{N}} \right] = \text{for positive anomalies} \tag{2}$$

$$RAI = -3 * \left[ \frac{N - \bar{N}}{\bar{X} - \bar{N}} \right] = \text{for negative anomalies} \tag{3}$$

Where N is monthly/yearly/seasonal rainfall (mm), $\bar{N}$ is average monthly/yearly/seasonal rainfall of the historical series (mm), $\bar{M}$ is average of the ten highest monthly/yearly/seasonal rainfall of the historic series (mm) and $\bar{X}$ is average of the ten lowest monthly/yearly/seasonal rainfall of the historic series (mm)

**2.2.3. Trend analysis.** *Mann-Kendall (MK) trend test.* To understand the long-term trends of rainfall and temperature, we applied the non-parametric Mann-Kendall (MK) test for monthly, seasonal, and annual time series data of temperature and rainfall from 1985 to 2020 in R statistical software (R4.1.3). The MK test is used to detect monotonically increasing or decreasing trends of annual and seasonal climate data [45]. The MK test is commonly used to identify trends in time series analysis due to its insensitivity to outliers and does not consider any distribution assumptions [46,47]. However, the result of the MK test may contain some error if autocorrelation exists in the time series data [48]. Therefore, in this study, autocorrelation is tested by calculating the autocorrelation coefficient at lag-1, and significant serial autocorrelation was found in the monthly rainfall records for December and June. Except for January, March and October monthly temperature data, significant serial autocorrelation was found in all months as well as average annual, average maximum and minimum temperature. Thus, to overcome this problem, the modified Mann-Kendall (MMK) method was used for serially autocorrelated data with a significant lag-1 autocorrelation coefficient using the variance correction method in R statistical software [49]. Based on [50,51] the MK statistics S is

computed using the following formula:

$$S = \sum_{i=1}^{n-1} \sum_{j=i+1}^{n} sign(x_j - x_i) \tag{4}$$

Where n is the number of data and xi and $x_j$ are sequential data values sign (.) is the sign function which can be calculated by the following equation.

$$sign(x_j - x_i) = \begin{cases} 1 \ if \ (x_j - x_i) > 0 \\ 0 \ if \ (x_j - x_i) = 0 \\ -1 \ if \ (x_j - x_i) < 0 \end{cases} \tag{5}$$

If the dataset is identically and independently distributed, then the mean of S is zero and the variance of S is given as:

$$var(s) = \frac{n(n-1)(2n+5) - \sum_{i=1}^{m} t_i \ (t_i - 1)(2t_i + 5)}{18} \tag{6}$$

Where m is the number of tied groups and $t_i$ is the number of data points in group t. When the number of sample size n>10, the standardized test statistic ($Z_{mk}$) is calculated as [50,51]:

$$Z_{mk=} \begin{cases} \dfrac{s-1}{\sqrt{var(s)}}, s > 0 \\ 0, \quad s = 0 \\ \dfrac{s-1}{\sqrt{var(s)}}, s < 0 \end{cases} \tag{7}$$

*Sen's slope estimator test.* The MK test does not provide an estimate of the magnitude of the trends [52]. Thus, in this study, we applied Theil-Sen approach, another nonparametric method, which is very popular among other techniques to quantify the slope of the trend or magnitude [53]. Sen's method has been widely used for determining trend magnitude in hydro-meteorological time series [54]. In this method, the slopes (β) of all data pairs are first calculated by using the following equation:

$$\beta = median\left(\frac{(x_j - x_k)}{(j - k)}\right) \tag{8}$$

for i = 1, 2,. . ., N, where $x_j$ *and* $x_k$ are data values at times j and k (j > k) respectively, and N is the number of all pairs $x_j$ and $x_k$. A positive value of $\beta$ indicates an upward (increasing) trend and a negative value indicates a downward (decreasing) trend in the time series.

*Innovative trend analysis method (ITA).* The ITA method, proposed by [55] has been used by several studies in combination with other trend analysis approaches to find differences in climatological, meteorological, and hydrological data time series due to its advantages over other non-parametric approaches [56]. ITA is valid regardless of the sample size, serial correlation structure of the time series, and non-normal probability distribution of the data [55]. In ITA, the hydro-meteorological time series were divided into two equal halves and then sorted both sub-series in ascending order. The first half of the series is placed on the X-axis, and the second half is placed on the Y-axis of the Cartesian coordinate system. If the data points on a scattered plot are collected on the 1:1 (45˚) straight line, it indicates there is no trend in the data. However, the trend is increasing when the data points fall above the 1:1 straight line and decreasing if the data points accumulate below the 1:1 straight line [55]. The indicator of trend is derived by dividing the mean difference between the linear line and the first half of the series.

The trend indicator of ITA is multiplied by 10 to make the scale similar to the other two tests (the Sen's slope estimator and MK tests). The trend indicator is calculated as:

$$D = \frac{1}{n} \sum_{i=1}^{n} \frac{10(x_j - x_i)}{\mu} \tag{9}$$

where D is trend indicator, n is number of observations in the subseries, $Xi$ is data series in the first half subseries class, $Xj$ is data series in the second half subseries class and $\mu$ is mean of data series in the first half subseries class. A positive and negative values of D indicate an increasing and decreasing trend, respectively. The ITA plots of annual and seasonal rainfall and temperature were generated using RStudio (package '*trendchange::innovetrend (X)*') [55].

**2.2.4. Pearson and partial correlation analysis.** In this study, temporal and spatial correlation between climate variables and key ecosystem goods and services such as food production, water supply, raw material provision, and climate regulation were examined. Since the distribution of the data affects parametric tests, histograms and normality tests were used to determine whether the distribution of the data was normal. Then, using the statistical package for the social sciences (SPSS) version 24, non-normally distributed data were transformed using Log10 data transformation techniques. The association between meteorological variables and key ecosystem services in 1985, 1995, 2005, 2015, and 2020 was evaluated using Pearson's correlation coefficient (r) with a two-tailed, 95% significance level. Ecosystem service values estimated from the LULC maps (S1 Fig; S1 and S2 Tables) were used to conduct the correlation analysis. Meteorological variables such as mean annual rainfall, main rainy season's rainfall, the second rainy season's rainfall, mean annual temperature, mean maximum temperature, mean minimum temperature, annual and seasonal potential evapotranspiration were included as climate variables in the correlation analysis. The correlation between key ecosystem services and mean annual rainfall, main rainy season's rainfall, the second rainy season's rainfall, mean annual temperature, mean maximum temperature, and mean minimum temperature was computed using Pearson's correlation coefficient. Pearson's correlation coefficient (r) is given by:

$$r = \frac{\sum_{i=1}^{n}(X_i - \bar{X})(Y_i - \bar{Y})}{\sqrt{\sum_{i=1}^{n}(X_i - \bar{X}) \sum (Y_i - \bar{Y})^2}}, \tag{10}$$

where n is the number of observations, x and y are the variables, and $\bar{X}$ and $\bar{Y}$ are the means respectively. The correlation coefficient (r) takes a value that ranges between +1 and −1, where a value of +1 represents a perfect positive correlation, −1 represents a perfect negative correlation, and a value of 0 indicates no correlation. The results of the Pearson correlation coefficient are categorized into several grading scales: very weak positive/negative correlation ($0 < |r| < \pm 0.2$); weak positive/negative correlation ($\pm 0.20 \leq |r| < \pm 0.4$), moderate positive/negative correlation ($\pm 0.40 \leq |r| < \pm 0.6$); strong positive/negative correlation ($\pm 0.60 \leq |r| < \pm 0.8$), and very strong positive/negative correlation ($\geq \pm .8$) following [57,58].

Potential evapotranspiration was computed using the Thornthwaite method [59] from monthly mean annual temperature data via the following equation:

$$PE = 16\left(\frac{10T}{I}\right)^a \tag{11}$$

where PE: monthly potential evapotranspiration, T: monthly mean air temperature (˚C), $I$: heat index, $a : 6.75 \; 10^{-7} I^3 - 7.71 \; 10^{-5} I^2 + -1.7921 \; 10^{-2} I + 0.49239$

The correlation between annual and seasonal evapotranspiration and key ecosystem services was calculated using the partial correlation analysis method in SPSS. The main rainy

season's rainfall, one of the main climatic determinants of food production in Ethiopia [60] was used as a control variable. Partial correlation coefficient is a more intrinsic correlation of the two variables which eliminates the influence of other variables on the correlation of these two variables [61] when the relationship between the variables was complex and influenced by multiple factors [62]. Partial correlation coefficient is calculated as:

$$r_{xy,z} = \frac{r_{xy} - r_{xz} * r_{yz}}{\sqrt{(1 - r^2_{xz}) * (1 - r^2_{yz})}} \tag{12}$$

where $r_{xy,z}$ is the partial correlation between variable $x$ and variable $y$ after independent variable $z$ is fixed, $r_{xy}$ is correlation between variable x and y, $r_{xz}$, is correlation of the third variable z with the variable x and $r_{yz}$ is correlation of the third variable z with the variable y.

The spatial dynamics of temperature, rainfall, and key ecosystem goods and services in the study area were analyzed by interpolating the annual average values of rainfall and temperature using the Inverse Distance Weighted (IDW) interpolation technique. In order to develop maps of key ecosystem goods and services, random points were generated for the study area and ecosystem service values of LULC classes were extracted using the 'extract point values' tool for each key ecosystem service included in this study. Then, using the IDW interpolation technique, a continuous surface was created for each ecosystem good and service, and their spatial association with meteorological variables was examined via the Pearson correlation coefficient. In addition, the average NDVI, which has a strong relationship with Net Primary Productivity (NPP), vegetation cover, and ecosystem productivity [63] for 1985, 1995, 2005, 2015, and 2020 were derived from the Landsat images (S1 Table). The spatial distribution and variation of NDVI in the study periods was analyzed using the NDVI maps made in ArcGIS and a Boxplot. The correlation between the NDVI and key ecosystem goods and services was examined using the Pearson correlation coefficient. NDVI, a measure of greenness is a reflectance recorded in the red and near-infrared band of the remote sensing imagery is calculated as:

$$NDVI = \frac{NIR - R}{NIR + R} \tag{13}$$

Where, R represents red and NIR represents near-infrared. Red = visible red Landsat band 3; NIR = near-infrared Landsat band 4 for Landsat TM, ETM+ and bands 4 and 5, respectively for OLI/TIRS. The NDVI value ranges from −1 to 1, where higher values indicate healthier and denser vegetation. Values lower than 0.1 represent bare areas of soil, rock, water (in the case of Maze) or snow elsewhere in other parts of the world snow occurs temporarily or permanently [64]. The overall methodological flow of the study is shown in Fig 3.

## 3. Results and discussion

### 3.1. Variability and trends of rainfall

The computed CV values of rainfall for all months were greater than 30%, ranging between 34.1% and 122.3% (Table 3). Therefore, there is high monthly rainfall variability in the study area, which is above 100% in December and January [43]. Whereas, rainfall exhibited moderate variability in the main (24.3%) and second (25.5%) rainy seasons and variability is low for annual rainfall (13.9%). Our findings indicate that monthly rainfall exhibits higher inter-annual variability than the mean annual and seasonal rainfall in the study area. This is consistent with other studies [65] which reported high monthly and moderate (spring) March-April-May rainfall variabilities in southern Ethiopia. However, CV value lower than 10% was

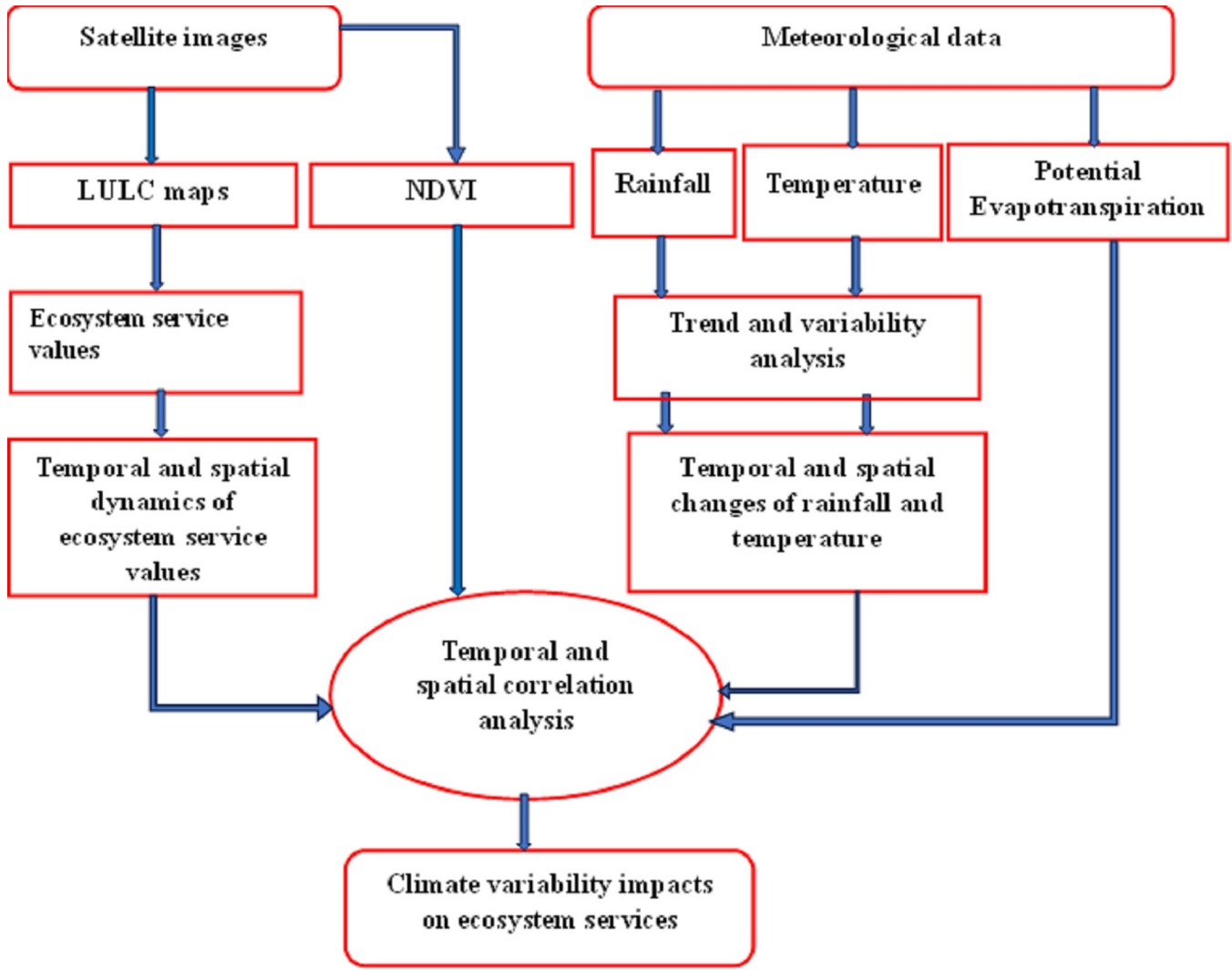

**Fig 3. Methodological framework of the study.**

reported in East Africa especially in western Uganda, Rwanda, Burundi, Tanzania and north-west Zambia with moist climatic conditions [66].

Similarly, the RAI of the annual and seasonal rainfall shows high inter-annual variability (Fig 4). Previous studies have shown that the country's rainfall variability is primarily driven by the seasonal shift of the Intertropical Convergence Zone (ITCZ), the influence of topography, and various interactions within the regional hydro-climate system [30,67]. The proportion of negative RAI values is 52.8%, 47.2%, and 52.8% in the mean annual, main rainy season, and second rainy season, respectively. This result implies that the study area experienced more dry periods than wet periods from 1985 to 2020. Negative annual and seasonal rainfall anomalies were also noted in the upper Genale river basin [68]. According to [69] RAI classification, 36.1% (annual), 38.9% (main rainy season), and 30.6% (second rainy season) of years were categorized as dry periods ranging from moderately dry to extremely dry conditions. Extremely dry conditions in the main rainy season and the mean annual rainfall were recorded in 1985, 1988, 1992, 1999, 2009, 2016, and 2017 (Fig 4). These periods were identified as drought periods in previous studies in Ethiopia [70], which coinciding with El Niño events [45].

**Table 3. Descriptive statistics, variability and MK trend test of monthly rainfall (1985–2020).**

| Months | Mean | Std. Deviation | CV (%) | MK Test (P-value) | Sen's slope |
|---|---|---|---|---|---|
| January | 24.371 | 26.332 | 108.0 | 0.131 | -0.364 |
| February | 29.312 | 22.636 | 77.2 | 0.487 | -0.301 |
| March | 80.2578 | 44.852 | 55.9 | 0.270 | -0.890 |
| April | 149.682 | 51.040 | 34.1 | 0.838 | -0.152 |
| May | 145.768 | 53.779 | 36.9 | 0.902 | 0.130 |
| June | 83.851 | 31.476 | 37.5 | 0.755 | -0.144 |
| July | 75.897 | 29.181 | 38.4 | 0.634 | -0.148 |
| August | 95.481 | 34.365 | 36.0 | 0.258 | 0.579 |
| September | 100.494 | 44.352 | 44.1 | 0.713 | 0.333 |
| October | 105.771 | 56.349 | 53.3 | 0.406 | 0.745 |
| November | 51.887 | 48.582 | 93.6 | 0.011** | 0.868 |
| December | 32.238 | 39.432 | 122.3 | 0.038 | -0.913 |
| Annual Rainfall | 975.009 | 135.394 | 13.9 | 0.649 | -0.721 |
| Main Rainy Season | 375.708 | 91.240 | 24.3 | 0.577 | -1.231 |
| Second Rainy Season | 301.748 | 76.833 | 25.5 | 0.186 | 1.901 |

** significant at 0.01 level.

**3.1.1. Mann Kendal's trend test of rainfall.** The MK test results (Table 3) indicated a decreasing trend of rainfall in December, January, February, March, April, June, and July. In addition, the mean annual, and the main rainy season rainfall also showed a decreasing trend. Unexpectedly, November rainfall alone showed a statistically significant increase at 99% confidence interval. The change point detection analysis conducted using the Pettitt's test [71] for November rainfall revealed that the year 2003 (p value 0.03) was a shift period for the month's rainfall. This might be explained by changes in the timing of the rainy seasons over the years, particularly since 2003, due to the impacts of climate change and variability. However, this requires further investigation. Even though the MK trend test results are statistically insignificant, the values of Sen's slope test and the slope value of the trend line equation (Table 3 and S2 Fig) show a high variability of rainfall in the study periods. This result implies that the monthly, annual, and seasonal rainfall in the study area does not have a monotonic increasing or decreasing trend, but it shows variability by fluctuating from its long-term mean (Table 3). Statistically non-significant increasing or decreasing trend of rainfall was reported by [72,73] in different agroecological zones of Ethiopia. The non-significant decreasing trend of the main rainy season (spring) rainfall in the study area is consistent with the findings at Negelle station [74] and across most parts of the country [60]. The decreasing trend of rainfall during the main rainy season affects crop production and food security [75]. Comparable to this finding, statistically non-significant changes in rainfall were observed in countries of the southern and eastern African regions, while statistically significant increases were recorded in the countries of the northern and central African regions [76].

**3.1.2. Innovative trend analysis of rainfall.** *The ITA statistical test result indicates a statistically insignificant decreasing trend in the main rainy season and mean annual rainfall. The ITA method is more sensitive in detecting hidden trends missed by the traditional MK tests [77] and hence showed a statistically significant (P<0.01) increasing trend for the second rainy season (autumn) rainfall (Table 4).*

Except for the second rainy season, it's all data points lie above the 1:1 line, the decreasing trend of the main rainy season and the mean annual rainfall can be seen from most of the

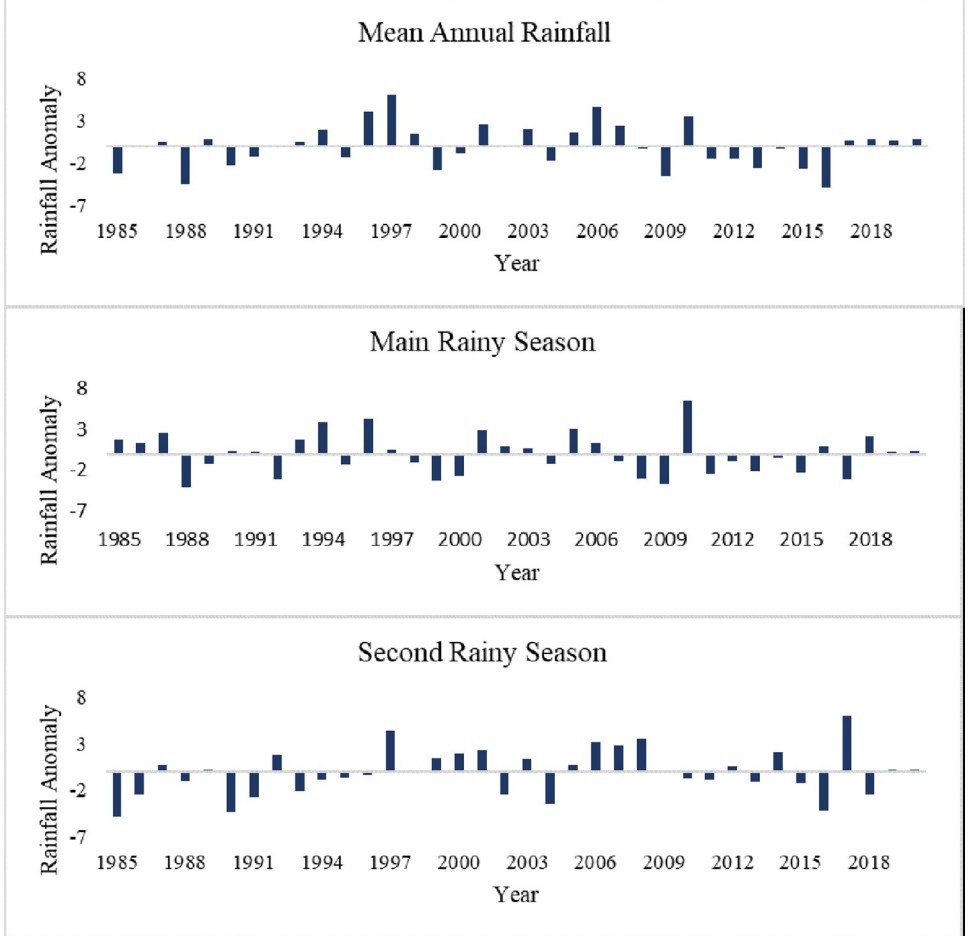

**Fig 4. Temporal variations in rainfall anomalies.**

scattered points that lie below the 1:1 line in the Cartesian coordinate system (Fig 5). From the ITA result of the mean annual rainfall, where the scatter points are closest around the 1:1 straight line, one can see that there is no significant trend.

## 3.2. Variability and trends of temperature

The mean annual temperature of the study area from 1985 to 2020 was 23.96˚C, and the mean maximum and mean minimum temperature were 31.18˚C and 16.75˚C, respectively, with the coefficients of variation 4.7%, 2.2%, and 12.6%, respectively (Table 5). The computed CV

**Table 4. *Innovative trend analysis of rainfall.***

| Seasonal and Annual RF | Trend Slope(s) | Trend Indicator(D) | UCL/LCL 95% | UCL/LCL 99% |
|---|---|---|---|---|
| Mean Annual RF | -0.492 | -0.090 | ±0.450 | ±0.591 |
| Main Rainy Season | -0.993 | -0.465 | ±0.580 | ±0.762 |
| Second rainy season | 1.487 | 0.928** | ±0.188 | ±0.247 |

** significant at the 0.01 level and UCL/LCL represent upper and lower confidence limits.

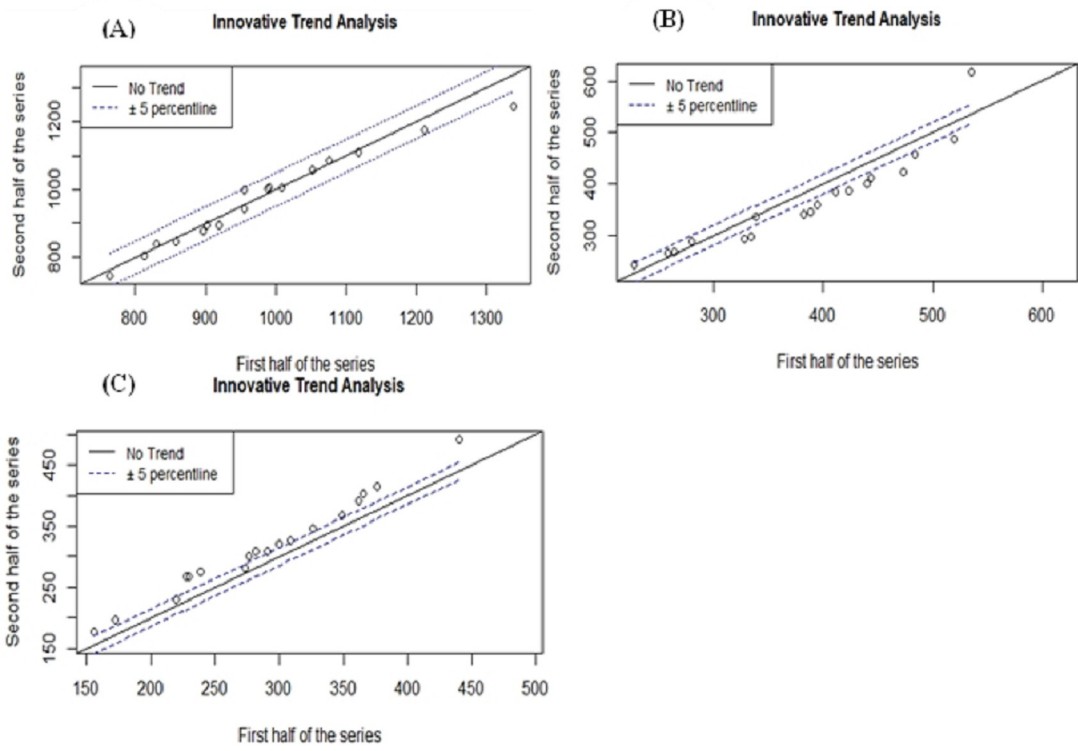

**Fig 5.** ITA of (A) mean annual rainfall, (B) main rainy season and (C) second rainy season.

values showed low variability of temperature for all months and 4.7%, 2.2%, and 12.6% for mean annual, maximum, and minimum temperature, respectively. This result shows that variability of the minimum temperature is the highest compared to the mean annual and maximum temperature. In general, variability of the mean annual, minimum, and maximum temperature was low, and the minimum temperature was more variable than the maximum temperature in the study area as well as in the country [78,79].

The inter-annual variability of the mean annual, maximum, and minimum temperature (Fig 6) indicates that the study area has experienced both warm and cool years from 1985 to 2020. The anomaly index exhibited a prolonged increase in the mean annual temperature from 2000 to 2014, and 1989 was the coolest year with -6.67 anomaly index value. The positive anomalies were indications of higher mean annual, maximum, and minimum temperatures than a long-term average, while the negative anomaly values indicate a lower temperature than the long-term mean. The higher proportion of positive anomaly values in the study area (55.6%, 63.9%, and 55.6% of the mean annual, minimum temperature and maximum temperature, respectively; Fig 6) implies that most of the study years were hotter than the long-term mean. This is most likely because of the changing climate, which leads to a rise in the country's temperature [80] in general and in southern Ethiopia [73] in particular.

**3.2.1 Mann Kendal trend test of temperature.** The MK test of monthly, mean annual, maximum, and minimum temperature showed no significant trend, where the computed p value was greater than the significance level. Even though the p value is greater than the significance level, the positive Sen's slope values of all months except for August, September, and November, exhibited the presence of an increasing trend in temperature (Table 5). In addition, the trend line equation (S3 Fig) also reveals that the slope of the trend line has a positive value,

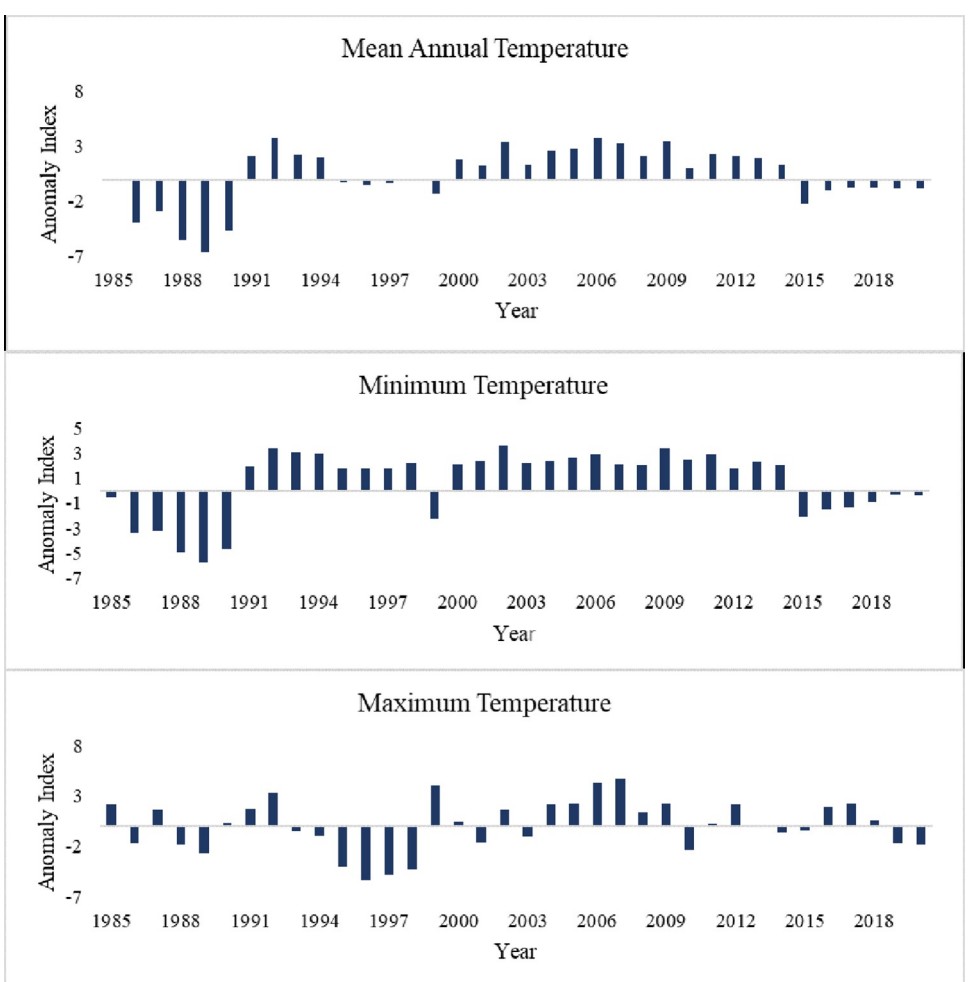

**Fig 6. Temporal variations in mean annual temperature anomalies.**

which implies an increasing trend of the mean annual, minimum, and maximum temperature. Similar results have been reported from eastern Kenya [81] and India [35], where the annual minimum and maximum temperatures show an increasing trend, whereas with variations in the monthly and seasonal minimum and maximum temperatures.

**3.2.2. Innovative trend analysis of temperature.** The statistical result of ITA (Table 6) shows significantly increasing trend in the mean annual, maximum, and minimum temperature at 0.01 significance level. A significantly increasing trend in the mean annual, maximum, and minimum temperature in Ethiopia was also reported by [82] at Lemi and Wereilu stations. In addition, from the graphical ITA result (Fig 7), one can notice that most of the scattered points fall above the 1:1 straight line, indicating an increasing trend in the mean annual, maximum, and minimum temperature in the study area.

## 3.3. Correlation between ecosystem services and climate variables

**3.3.1. Spatial dynamics of ecosystem services in relation to climate variables and NDVI.** We tested spatial correlation between the values of key ecosystem services (food production, water supply, raw material provision, and climate regulation) and mean annual rainfall, and mean annual temperature. The mean annual rainfall from 1985–2020 shows high

**Table 5. *Descriptive statistics, variability and MK trend test of monthly temperature (1985–2020).***

| Months | Mean | Std. Deviation | CV (%) | MK Test (P-value) | Sen's slope |
|---|---|---|---|---|---|
| Jan | 24.578 | 1.528 | 6.2 | 0.191 | 0.031 |
| Feb | 25.397 | 1.670 | 6.6 | 0.233 | 0.046 |
| Mar | 25.249 | 1.716 | 6.8 | 0.141 | 0.044 |
| Apr | 24.117 | 1.478 | 6.1 | 0.140 | 0.030 |
| May | 23.732 | 1.310 | 5.5 | 0.712 | 0.008 |
| Jun | 23.457 | 1.494 | 6.4 | 0.820 | 0.002 |
| Jul | 23.063 | 1.384 | 6.0 | 0.798 | 0.006 |
| Aug | 22.974 | 1.325 | 5.8 | 0.865 | -0.005 |
| Sep | 23.32 | 1.331 | 5.7 | 0.733 | -0.011 |
| Oct | 23.521 | 1.111 | 4.7 | 0.978 | 0.002 |
| Nov | 23.859 | 1.130 | 4.7 | 0.733 | -0.013 |
| Dec | 24.295 | 1.361 | 5.6 | 1.000 | 0.003 |
| Mean Annual | 23.964 | 1.136 | 4.7 | 0.887 | 0.001 |
| Maximum Temp | 31.178 | .672 | 2.2 | 0.394 | 0.010 |
| Minimum Temp | 16.748 | 2.110 | 12.6 | 0.532 | 0.007 |

spatial variability in the study area, ranging from 830.79 mm to 1044.01 mm (Fig 8). A large proportion of the study area experiences lower rainfall except in the northeastern part, with the highest mean annual rainfall value of about 1044.01 mm. The spatial distribution of the mean annual rainfall increases from the southeastern to northeastern parts of the study area. Whereas, almost all parts of the study area receive higher temperature except the northeastern part, which has the lowest temperature, about 22.07°C. The spatial distribution of the mean annual temperature increases from the northeastern part to the central and southeastern parts. The highest mean annual rainfall values were observed in the highest elevation areas, and the lowest mean annual rainfall values were observed in the lowest elevation areas. Conversely, the lowest mean annual temperature values were observed in the highest elevation areas (Fig 8). This indicates that the spatial distribution of the mean annual rainfall and temperature was related to the topography of the study area. Studies also reported that rainfall distribution is highly correlated with topography in Eastern Ethiopian highlands [75] and Meki watershed in the rift valley region [83].

The spatial correlation analysis result (Table 7) indicates that water supply, raw material provision, and climate regulation services have a significant positive correlation with the mean annual rainfall. In agreement with this finding, positive spatial correlation between precipitation and regulatory services was found in northern China [84]. Whereas, all ecosystem services included in this study showed a negative correlation with the mean annual temperature. These indicate that areas with higher rainfall have better food production, water supply, raw material, and climate regulation service provision than areas receiving lower rainfall. On the other

**Table 6. *Innovative trend analysis of temperature.***

| Temperature | Trend Slope(s) | Trend Indicator(D) | UCL/LCL 95% | UCL/LCL 99% |
|---|---|---|---|---|
| Mean Annual Temp | 0.042 | 0.319** | ±0.006 | ±0.007 |
| Max. Temp | 0.024 | 0.139** | ±0.002 | ±0.003 |
| Min. Temp | 0.060 | 0.662** | ±0.005 | ±0.007 |

** significant at the 0.01 level and UCL/LCL represent upper and lower confidence limits.

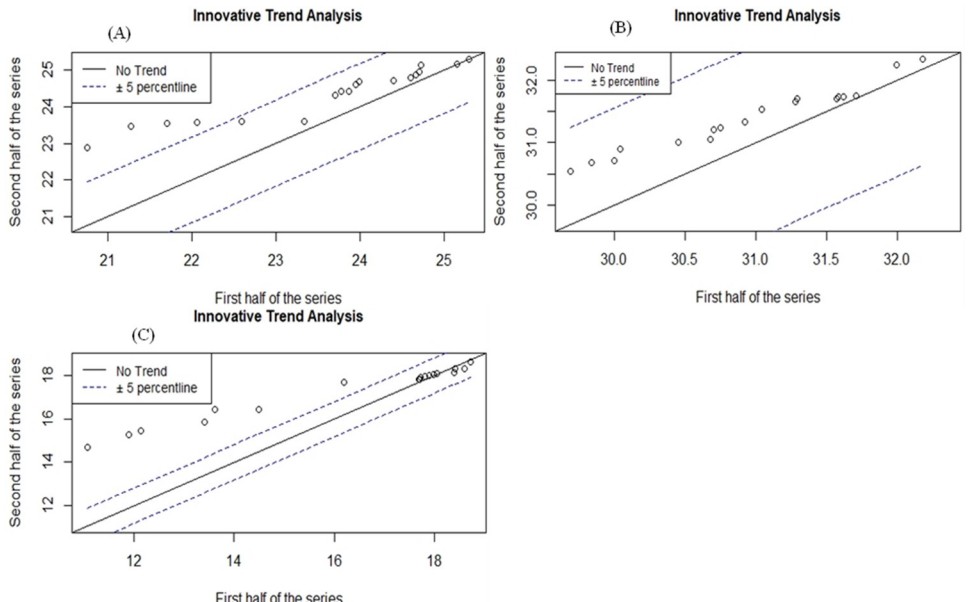

**Fig 7.** ITA of (A) mean annual temperature, (B) maximum temperature and (C) minimum temperature.

hand, areas with high mean annual temperature have a lower provision of ecosystem services and need management measures to reduce the negative impacts of temperature increase on ecosystem services. Therefore, the correlation result (Table 7) implies that the spatial variability of rainfall affects the provision of key ecosystem services essential for human wellbeing. Because changes in rainfall patterns increase the frequency of climate extreme events and have an effect on ecosystem services [80].

The spatial distribution of NDVI, a measure of greenness that is highly correlated with NPP, plant cover, and ecosystem productivity [63], in 1985, 1995, 2005, 2015, and 2020 and the mean NDVI were calculated and mapped for the study area (Fig 9). The spatial distribution of the annual and mean NDVI in the study area varies widely, ranging from -0.14 to 0.68, -0.11 to 0.74, -0.37 to 0.50, -0.01 to 0.53, 0.00 to 0.57, and -0.22 to 0.63 in 1985, 1995, 2005, 2015, 2020, and average NDVI, respectively (Fig 9). The highest NDVI for all study periods was primarily observed in the northern and southwestern parts of the study area, with varying amounts. As elevation is found to be the most important topographic factor that determines the spatial distribution of NDVI [84], the northern and southwestern parts of the study area have high elevation and thus high NDVI values. Additionally, the Maze stream course, which includes riverine forests, also has a higher NDVI. The spatial distribution of the NDVI was linked to land cover types [62] with higher NDVI values found in riverine forests, wooded grasslands, and cropped farmlands. The lowest NDVI values were observed primarily in bare lands, fallowed farmlands, and built-up areas (Figs 9 and S1).

Fig 10 shows the inter-annual variability of NDVI from 1985–2020 in boxplots. The boxplot, which shows a significant variation among the study periods, is developed from the NDVI data of 1985, 1995, 2005, 2015, and 2020. The minimum value of NDVI is -0.37, which was in 2005, and the maximum one is 0.74, which was in 1995. In addition, the 25th percentile of the NDVI ranges from ~ - 0.1 to ~ 0.2 while the 75th percentile of the NDVI ranges from ~ 0.1 to ~ 0.5 across the study periods. Based on the boxplots (Fig 10), NDVI increased from 1985 to 1995 and from 2005 to 2015 and 2020. From 1995 to 2005, a significant decrease in NDVI was observed. A decrease in NDVI from 1995 to 2005 was mostly due to people and

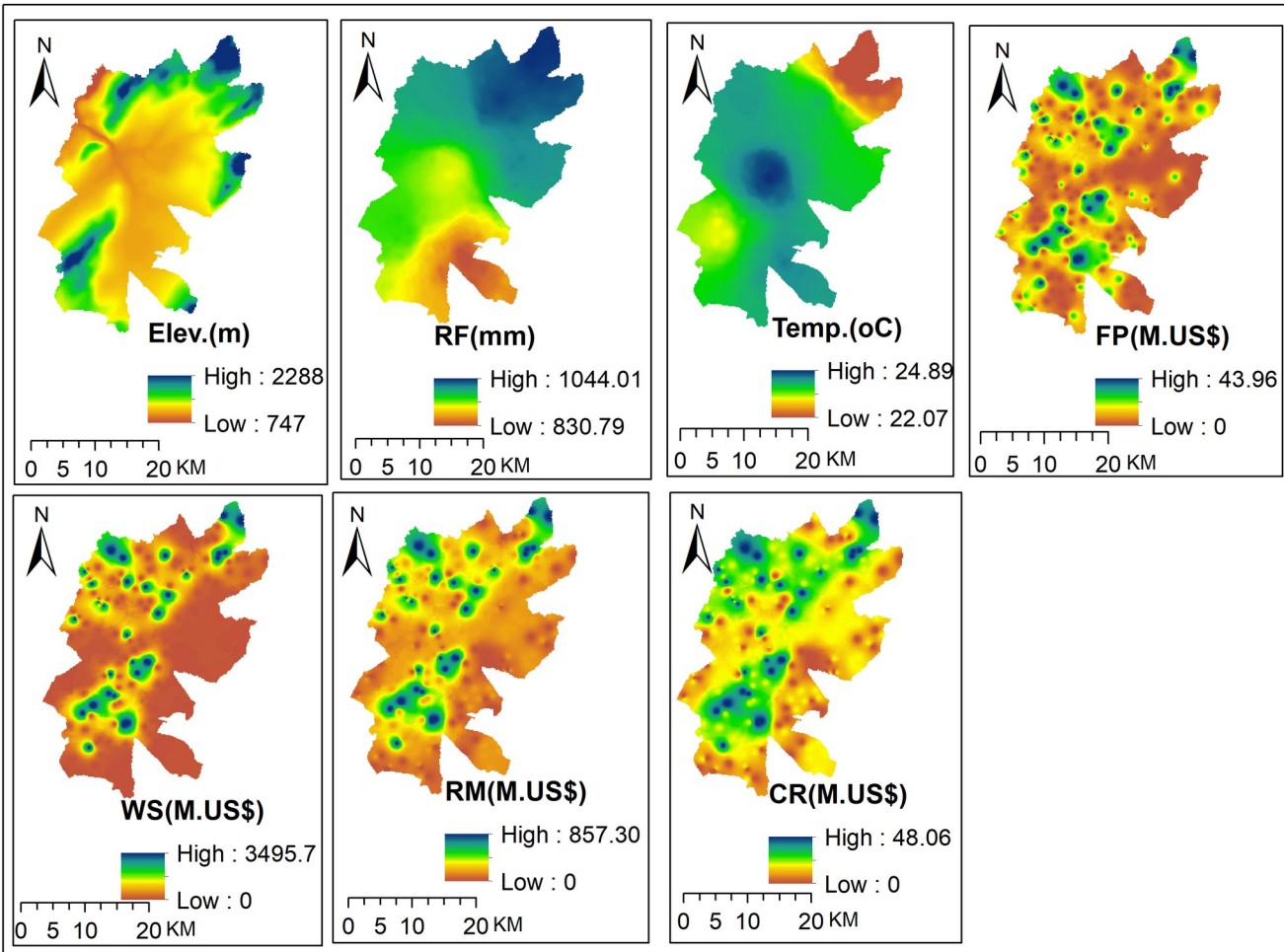

**Fig 8. Spatial distribution of ecosystem services and climate variables.** Note: Elev. = Elevation, RF = Rainfall, Temp. = Temperature, FP = Food Production, WS = Water Supply, RM = Raw Material, and CR = Climate Regulation.

domestic animals unrestricted access and effect on vegetation coverage of the study area by cutting grass for their cattle and for thatching roofs, and cutting trees for construction material and firewood. The establishment of national parks contributes to maintaining the ecosystem

**Table 7.** *Spatial correlation among ecosystem services, climate variables and NDVI.*

| Ecosystem Services | Pearson Correlation | Mean Annual Rainfall | Mean Annual Temperature | NDVI |
|---|---|---|---|---|
| Food production | r | .176 | -.206 | .051** |
|  | Sig. | .136 | .081 | .000 |
| Water Supply | r | .268* | -.168 | .111** |
|  | Sig. | .022 | .156 | .000 |
| Raw material provision | r | .268* | -.168 | .126** |
|  | Sig. | .022 | .156 | .000 |
| Climate Regulation | r | .269* | -.167 | .166** |
|  | Sig. | .021 | .158 | .000 |

**. Correlation is significant at the 0.01 level (2-tailed).

*. Correlation is significant at the 0.05 level (2-tailed).

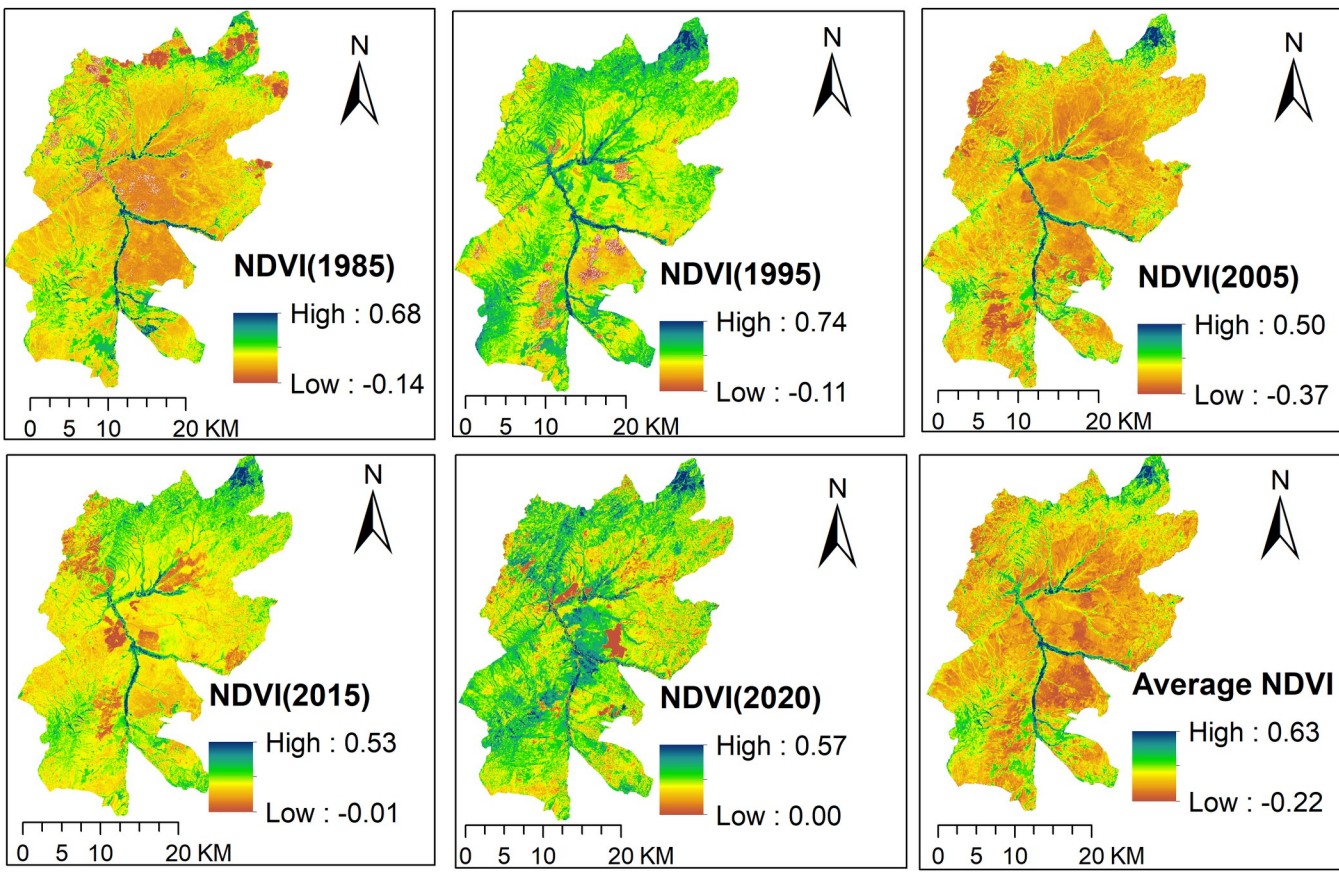

**Fig 9. Spatial distribution of NDVI from 1985 to 2020.**

and biodiversity conservation [15]. Maze national park was established in 2005, and since then, unrestricted access to park resources has been prohibited, allowing degraded fields to regenerate grasses and trees. This led to gradual increase in NDVI after 2005, especially inside the park boundary (Fig 10).

We carried out a correlation analysis between the mean NDVI (the average of 1985, 1995, 2005, 2015, and 2020) and key ecosystem services. The Pearson correlation analysis result showed a significant positive correlation between NDVI and all key ecosystem services (Table 7). This implies that areas with high elevation, high rainfall and relatively dense vegetation coverage or NDVI (Fig 8) provide better food production, water supply, raw material, and climate regulation services. Positive correlations between NDVI and ecosystem services, for instance, water yield, soil conservation, and carbon storage services, have also been reported in southern China [85]. In the same vein, NDVI showed positive correlation with NPP and overall ecosystem productivity in India [63].

**3.3.2. Temporal correlation between climate variables and ecosystem services.** Temporal correlation between the values of key ecosystem services and rainfall (mean annual, main rainy season, and secondary rainy season), temperature (mean annual, maximum, and minimum temperature), and potential evapotranspiration (the annual, main rainy season and second rainy season) was analyzed using the Pearson's correlation coefficient at 95% confidence level.

Food production had a positive (r = 0.475) correlation with the main rainy season's rainfall than the mean annual rainfall and the second rainy season's rainfall (Table 8). This positive

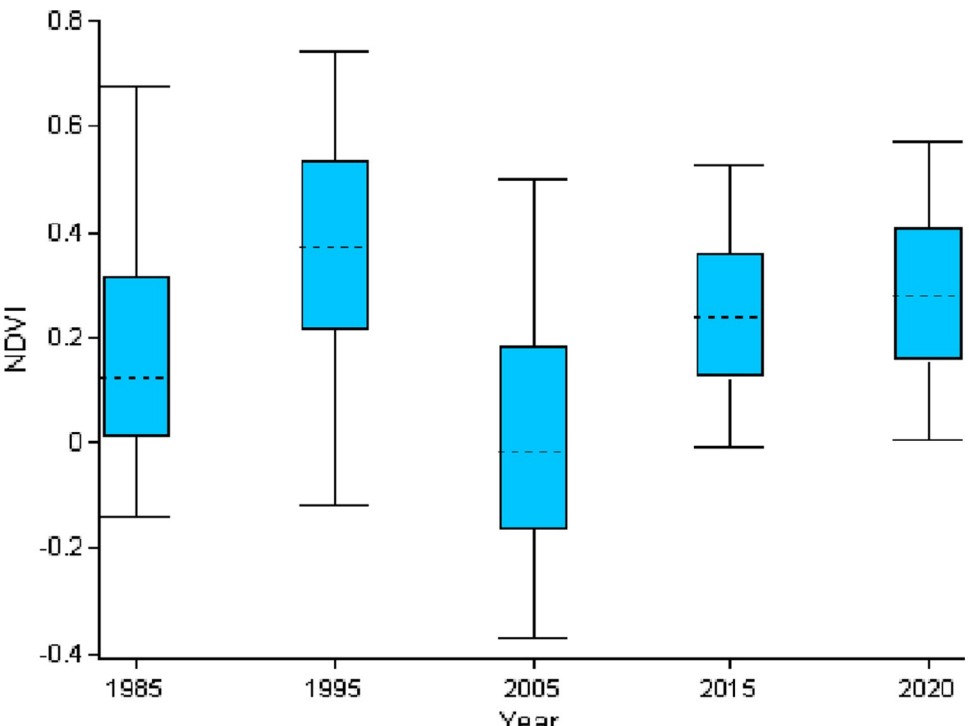

**Fig 10. Boxplot showing the NDVI's inter-annual fluctuation from 1985–2020.**

statistical correlation coefficient suggests that higher precipitation is linked to higher food production services. Since rainfall is one of the main climatic determinants of food production in Ethiopia [60], wetter years are generally associated with higher food production, while dry years are linked to lower production. Therefore, despite the fact that the correlation is not statistically significant, rainfall variability has an impact on food production in the study area. The second rainy season (autumn) rainfall showed an increasing trend in the study area (Table 3) and exhibited moderate negative correlation (r = -.496; Table 8) with food production service. This result reveals that an increase in rainfall might not be an indication of improved agricultural practices and food production since an increasing trend of rainfall would not necessarily mean good distribution of rainfall, especially during the crop-growing season [86].Whereas, the correlation analysis showed a strong negative correlation (r = -.837;

**Table 8. Temporal correlation between ecosystem services and climate variables (1985–2020).**

| Ecosystem Services | Pearson Correlation | Mean Annual RF | MRS RF | SRS RF | Annual Temp. | $T_{max}$ | $T_{min}$ |
|---|---|---|---|---|---|---|---|
| Food production | r | .247 | .475 | -.496 | .800 | -.349 | -.837 |
|  | Sig. | .689 | .419 | .396 | .104 | .565 | .077 |
| Water supply | r | -.568 | .207 | .512 | .248 | -.196 | -.115 |
|  | Sig. | .318 | .738 | .377 | .688 | .752 | .854 |
| Raw material provision | r | -.494 | .291 | .448 | .326 | -.234 | -.178 |
|  | Sig. | .398 | .635 | .450 | .592 | .704 | .775 |
| Climate regulation | r | -.428 | .350 | .394 | .385 | -.252 | -.231 |
|  | Sig. | .472 | .563 | .512 | .522 | .683 | .709 |

Note: MRS, Main rainy season; SRS, second rainy season; RF, Rainfall; $T_{max}$, Maximum Temperature; $T_{min,}$ Minimum Temperature.

Table 8), with minimum temperature and moderate negative correlation with maximum temperature. This is due to decreasing crop productivity for even small local temperature increases in arid and semi-arid areas [6].

The quantity or availability of water for various purposes is very much dependent on the amount of rain, and annual and seasonal rainfall variability significantly affects rural water supply services [87]. Likewise, water supply service in the study area had a positive correlation with the main and second rainy season's rainfall (Table 8). This result implies that, since precipitation affects the availability of water, the main and second rain seasons positively contribute to water availability and supply in the study area. On the other hand, the water supply service showed a negative relationship with maximum and minimum temperature, with a correlation coefficient of (r = -.196 and -.115, respectively; Table 8). This result indicates that an increase in minimum and maximum temperature exerts a negative impact and reduces the supply of water in the study area, though the correlation is not significant. Similarly, previous studies reported climate variability and increasing temperature reduced the availability of water in the semi-arid lowlands [6] and in sub–Saharan Africa [88].

Raw material provisioning services include the production of lumber, fuelwood, or fodder that are extracted as raw materials [36]. In the local context, raw materials also encompass the production of charcoal, thatching grass, pasture, and cutting trees for building houses and fences. As indicated in Table 8, raw material provisioning service exhibited a positive correlation with the main and second rainy season's rainfall (Table 8). But a negative correlation was observed with the maximum and minimum temperature (r = -.234 and -.178, respectively; Table 8). This result suggests that the reduction of raw material provisioning services in the study area is attributed to the rising impact of temperature along other determining factors. Rising temperature and altered precipitation patterns would increase aridity, worsen water stress, alter the distribution pattern of grassland in the tropical grassland ecosystem, and consequently reduce raw material provisioning services [7]. In line with this finding, a study in Kenya noted that climate change and variability, such as recurrent drought and rainfall variability, along with other anthropogenic factors, are reported as determining factors of provision of pasture, thatching grass, and fuelwood for household energy needs including other ecosystem services [89].

The climate regulation service and the main and second rainy seasons' rainfall showed a positive correlation, though the correlation coefficient is low (Table 8). This positive correlation resulted from the increased precipitation, which promotes an increase in vegetation density [84] and hence contributes positively to climate regulation. A similar positive correlation between precipitation and regulatory services was reported by [84,90]. On the other hand, the climate regulation service in the study area exhibited a negative correlation with the minimum and maximum temperature (Table 8). A rise in both minimum and maximum temperatures exert pressure on vegetation coverage and climate regulation services. A previous study in the Sudan identified an increase in minimum and maximum temperature and decreasing precipitation as a threat to vegetation coverage of the country [91]. In addition to climate change and variability, human activities such as LULC changes have also contributed to the decline in vegetation cover and the reduction of climate regulation services in the study area. This is due to the encroachment of farmlands and built-up areas over naturally vegetated lands [33,34].

The trend analysis results (Tables 3–6) illustrated increasing trends in the mean annual, maximum, and minimum temperature, whereas the mean annual and main rainy season's rainfall showed decreasing trends from 1985 to 2020 in the study area. Increasing temperature resulted in water loss due to evaporation, and this affected agriculture and livestock production, domestic water supply, and municipal services in southern Ethiopia [45] and in the tropics [86]. Similarly, the correlation analysis result exhibited a negative relationship between

**Table 9. *Partial correlation between potential evapotranspiration and ecosystem services.***

| Ecosystem Services | Partial Correlation | Annual PET | MRS PET | SRS PET |
|---|---|---|---|---|
| Food production | r | -.929 | -.990 | -.814 |
| | Sig. | .071 | .010 | .186 |
| Water supply | r | -.017 | .135 | -.145 |
| | Sig. | .983 | .865 | .855 |
| Raw material provision | r | -.005 | .140 | -.162 |
| | Sig. | .995 | .860 | .838 |
| Climate regulation | r | -.012 | .122 | -.210 |
| | Sig. | .988 | .878 | .790 |

Note: MRS, Main rainy season; SRS, second rainy season; PET, potential evapotranspiration.

maximum and minimum temperature and key ecosystem services, though the correlation was not statistically significant. A positive trend of temperature (Tables 5 and 6) shows that there has been a rising sign of climate change in the study area and closely similar results have been reported in Kenya [92] that negatively affects the provisioning as well as regulation services of ecosystems at local and regional scales in varying degrees. The main rainy season is the major contributor to the annual rainfall in the study area, and during this season, food production, water supply, and vegetation growth are expected to be better once the season turns out. As indicated in Table 8, main rain season's rainfall exhibited a positive relationship with all ecosystem services included in this study. Previous studies in China and the 2014 World Food Program (WFP) report shows a positive correlation between precipitation and different provisioning and regulatory ecosystem services, for instance, Water yield, NPP, and soil conservation services [60,84,90].

The relationship between annual and seasonal evapotranspiration and key ecosystem services was explored using the partial correlation analysis. Main rainy season rainfall was used as a control variable since rainfall is one of the main climatic determinants of food production in Ethiopia [60] and greatly affects provisioning of ecosystem services. The annual and seasonal potential evapotranspiration exhibited a strong negative correlation with food production service (r = -.929, -.990, and -.814 for the annual, main rainy season, and second rainy season, respectively; Table 9). Water supply, raw material provisioning, and climate regulation services also showed a similar negative correlation with annual and seasonal potential evapotranspiration (Table 9). Evapotranspiration greatly contributes to the water loss [93], which has a negative impact on the water resources and water supply [94]. Therefore, insufficient water supply affects the growth of crops and harvests, and reduces food supply [94]. Particularly, the impact is strong in arid and semi-arid regions where a small amount of precipitation is available for plant growth [95]. Comparing the relationship between key ecosystem services and climate variables, the negative impact of both annual and seasonal evapotranspiration on food production was the strongest. In arid regions, a large proportion of precipitation is returned to the atmosphere through evapotranspiration [95] due to increased temperature [45], affecting soil moisture availability and, consequently, the provision of key ecosystem services.

## 4. Conclusion

Our study demonstrated statistically significant positive spatial correlation among key ecosystem services and the mean annual rainfall and NDVI, whereas, negative correlations were observed among key ecosystem services and mean annual temperature. Temporally, key ecosystem services and main rainy season rainfall were positively correlated. However, negative

correlations were observed among ecosystem services and maximum and minimum temperatures. In addition, a strong negative correlation was observed between food production service and potential evapotranspiration. Therefore, the spatiotemporal variability of rainfall and temperature has largely negative effect on the capacity of the ecosystem to provide services that are essential for human well-being. Employing various statistical methods and extensive data, the patterns observed in our study are not confined to MzNP and its surroundings but are indicative of broader trends in similar ecosystems globally. Thus, our findings could be applicable to other semi-arid regions experiencing similar pressures from LULC changes and climate variability.

Understanding the levels of severity of climate variability is the first and fundamental step in planning and implementing appropriate measures. Hence, designing ecosystem conservation and restoration strategies such as implementing reforestation and afforestation, and balancing conservation and agriculture through integrated land management are crucial. Such strategies require involving the local communities in conservation planning and decision-making process, and raising awareness of the local community about the importance of ecosystem conservation. Thus, ecosystem conservation and restoration are crucial steps to increase the potential of ecosystems to adapt and mitigate the impacts of climate change and variability. We considered only four ecosystem services, which may have limitations to fully representing the status of the study region's ecosystem services. Additionally, because there was a dearth of documented historical and field data in the research area, the quantification of ecosystem service values was based solely on the dynamics of LULC classes across space and time. Therefore, we recommend future research in this and other similar settings, such as national parks under pressure from human activities and climate variability, to take into account other important ecosystem services. Additionally, it is important to estimate the values of these services using more precise models (INVEST), market price-based valuation methods, and extensive field data.

## Supporting information

**S1 Fig. Land use land cover maps.**
(TIF)

**S2 Fig. Trends of annual rainfall, main rainy season and second rainy season (1985–2020).**
(TIF)

**S3 Fig. Trends of mean, maximum and minimum temperature (1985–2020).**
(TIF)

**S1 Table. Satellite images used for LULC and NDVI analysis.**
(DOCX)

**S2 Table. Accuracy assessment result of the classified images.**
(DOCX)

**S1 Dataset. Monthly average rainfall, maximum, and minimum temperature (1985–2020).**
(XLSX)

**S2 Dataset. Rainfall data (1985–2020).**
(XLSX)

**S3 Dataset. Maximum temperature (1985–2020).**
(XLSX)

**S4 Dataset. Minimum temperature (1985–2020).**
(XLSX)

**S5 Dataset. Points extracted from image.**
(RAR)

## Acknowledgments

We acknowledge Hawassa University for providing a study leave. We are also grateful to the Meteorological Services Agency, Southern district office, for providing us with meteorological data.

## Author Contributions

**Conceptualization:** Mestewat Simeon, Desalegn Wana.

**Data curation:** Mestewat Simeon.

**Funding acquisition:** Desalegn Wana, Zerihun Woldu.

**Investigation:** Desalegn Wana.

**Methodology:** Mestewat Simeon, Desalegn Wana.

**Project administration:** Desalegn Wana, Zerihun Woldu.

**Resources:** Desalegn Wana.

**Software:** Mestewat Simeon.

**Supervision:** Desalegn Wana, Zerihun Woldu.

**Validation:** Zerihun Woldu.

**Writing – original draft:** Mestewat Simeon.

**Writing – review & editing:** Mestewat Simeon, Desalegn Wana, Zerihun Woldu.

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
