## [Decision Letter · Decision Letter 0]

5 Jun 2024

PONE-D-24-05650Spatiotemporal Dynamics of Ecosystem Services in Response to Climate Variability in Maze National Park and its Environs, Southwestern EthiopiaPLOS ONE

Dear Dr. Simeon,

Thank you for submitting your manuscript to PLOS ONE. After careful consideration, we feel that it has merit but does not fully meet PLOS ONE’s publication criteria as it currently stands. Therefore, we invite you to submit a revised version of the manuscript that addresses the points raised during the review process.

Two experts have recommended accepting your manuscript for publication, pending major changes. Please note that both reviewers raised relevant concerns regarding the structure of the manuscript and agreed on the lack of detail in the Methods section. They also recommend putting your contribution into a broader perspective. Since the changes required are too substantive, I am willing to consider a revised version for publication in this journal, assuming you modify the manuscript according to all recommendations. 

We look forward to receiving your revised manuscript.

Kind regards,

Angelina Martínez-Yrízar, Ph.D.

Academic Editor

PLOS ONE

Journal Requirements:

4. We note that your Data Availability Statement is currently as follows: [All relevant data are within the manuscript and its Supporting Information files]

5. Please upload a copy of S1 Figure, S2 Figure, S3 Figure and S1 Table to which you refer in your text on page 33. Please amend the file type to 'Supporting Information'. If the Supplementary file is no longer to be included as part of the submission please remove all reference to it within the text.

Reviewers' comments:

Reviewer's Responses to Questions

**Comments to the Author**

1. Is the manuscript technically sound, and do the data support the conclusions?

Reviewer #1: Partly

Reviewer #2: Yes

2. Has the statistical analysis been performed appropriately and rigorously? 

Reviewer #1: Yes

Reviewer #2: Yes

3. Have the authors made all data underlying the findings in their manuscript fully available?

Reviewer #1: Yes

Reviewer #2: Yes

4. Is the manuscript presented in an intelligible fashion and written in standard English?

Reviewer #1: Yes

Reviewer #2: Yes

5. Review Comments to the Author

Reviewer #1: The research titled “Spatiotemporal Dynamics of Ecosystem Services in Response to Climate Variability in Maze National Park and its Environs, Southwestern Ethiopia” aims to investigate the impacts of climate variability on selected ecosystem services in Maze National Park and its surroundings areas. The study utilizes various statistical methods, including the Mann-Kendall (MK) trend tests, Sen’s slope estimator, and innovative trend analysis (ITA), to assess climate trends and their effects on ecosystem services. The research focuses on understanding the relationships between climate variables (e.g., rainfall and temperature) and ecosystem services using correlation analyses.

The manuscript demonstrates a solid foundation and makes a valuable contribution to the fields of ecosystem services and climate variability. The research is well-conceived, and the methods are appropriately chosen to address the study’s objectives. However, the manuscript in its current form may not meet the standards of PLOS ONE due to the areas needing improvement that I describe below.

Overall, the coherence between the introduction, methods, results, and discussion is strong. Each section logically follows the previous one, maintaining a clear narrative throughout the manuscript. I suggest ensuring that the terminology and definitions used are consistent throughout the manuscript to avoid confusion.

Abstract:

The abstract summarizes the study effectively, but lacks specific quantitative findings.

Introduction:

The introduction provides a comprehensive background on the importance of ecosystem services and the impact of climate variability. It effectively highlights the relevance of the study by emphasizing the lack of empirical research on this topic in Ethiopia. The introduction outlines the specific objectives of the study, aiming to fill this research gap, by focusing on the spatiotemporal impacts of climate variability on key ecosystem services in MzNP. However, there are some critical points to be improved:

1. The references are appropriate but could be expanded to include more recent studies on similar topics globally to strengthen the argument for the study’s novelty.

2. The identification of research gap is well articulated, but it could benefit from more detailed discussion on why previous studies were insufficient.

3. The text should consistently distinguish between different types of ecosystem services to enhance clarity.

2. It is necessary to use consistent terms for climate-related phenomena and impacts. For example, “climate change” and “climate variability” are used interchangeably but have distinct meanings.

3. To give some examples of the mechanisms by which climate change affects ecosystem services, will provide a more comprehensive understanding of the linkages and processes involved.

4. The aims are quite broad, and the authors don’t provide specific research questions that guide the research. Defined aims and question would provide a sharper focus for the study.

Methods:

The methods section is detailed and describes the study area, however, discuss specific characteristics of MzNP that make it a relevant case study for examining the impacts of climate variability on ecosystem services, such as its biodiversity, ecosystem types, or known climate vulnerabilities.

The methods for collecting climate data are clear, but the explanation for ecosystem services data is vague. I suggest providing a more detailed explanation of how ecosystem services data were measured or quantified. I suggest specifying the indicators used, the data sources, and the methods of data collection.

The use of the MK trend test, Sen’s slope estimator, and ITA are appropriate for analyzing climate trends. The use of Pearson correlation and partial correlation to evaluate relationships among variables is also well-justified. However, while the description of the study area is thorough, including more specific reasons for choosing MzNP would strengthen the rationale. Finally, there should be a discussion on the assumptions underlying the statistical methods and how they were addressed, this will provide robustness to the methodology and ensure the validity of the results.

Results:

The results are presented clearly, with the findings from the MK test, ITA and correlation analyses well-organized. The results show significant trends in climate variables and their correlations with ecosystem services. However, visual aids (e.g, graphs, charts) could enhance the clarity and impact of the findings. While the results are clear, the interpretation could be more nuanced. I suggest a time series graph showing temperature and rainfall trends over the study period would provide a clearer picture of the data.

Discussion:

The discussion section interprets the results in the context of existing literature, emphasizing the implications for ecosystem services and climate adaptation strategies. It addresses the study’s limitations and suggests areas for future research. However, while the results are clear, the interpretation could be more nuanced. I suggest discussing potential reasons for the observed trends and correlations would add depth. This could provide a broader context for results. I suggest discussing potential ecological or social factors that might explain the observed trends and correlations. For example, consider the role of land-use changes, conservation policies, or socio-economic conditions in the region.

There is limited comparison with findings from other regions. This will help place the results in a broader context and highlight similarities or differences in climate impacts on ecosystem services across different regions.

In general, the discussion is insightful but could benefit from deeper analysis. I suggest exploring the mechanisms driving the observed trends and correlations in greater detail. For example, discuss how specific climatic factors, such as seasonal rainfall patterns or temperature extreme, affect different ecosystem services like water provisioning, biodiversity, or carbon sequestration.

On the other hand, the practical implications are mentioned but not elaborated upon. I suggest providing concrete recommendations for local stakeholders, such as adaptive management practices, conservation strategies, or community engagement approaches to mitigate the impacts of climate variability. I suggest providing specific recommendations for future research methodologies and potential study areas.

I suggest integrating more recent studies and broader geographical contexts to enhance the discussion. This will help situate the study within the larger body of research on climate variability and ecosystem services.

Reviewer #2: GENERAL COMMENTS

In the article titled “Spatiotemporal Dynamics of Ecosystem Services in Response to Climate Variability in Maze National Park and its Environs, Southwestern Ethiopia”, the authors evaluated the impacts of temperature and rainfall variability on selected key provisioning (food production, water supply, raw material) and regulatory ecosystem services (climate regulation) in Maze National Park.

In this regard, I comment that the introduction is too general, it only mentions the trends or response patterns of ecosystem services to the effect of climate change or climate variability, which are not necessarily synonyms. It is necessary to provide greater context of the effect on ecosystem services, explain response mechanisms, processes involved, etc. Furthermore, it is not clearly justified why the four key services were selected. Services are not defined. For food production and raw material services, it would be appropriate to clearly define which crops or raw materials are being considered within the context of Maze Nation Park. It is recommended to show more forcefully how this case study contributes to the global literature, not just the country or the national park, on the response of ecosystem services to climate variability when considering a wide range of data (years, precipitation, temperature). and analysis techniques (CV, RAI, MK, Sen slope, ITA, Pearson and partial Correlation Analysis).

In the methods, there is an imbalance in describing how the data was processed. For the climatic variables there is very fine detail in how they were processed and the tests used to show trends and variability are clearly indicated. On the other hand, for services, it is only indicated that LULC maps are used and transformed into four ecosystem services. It is not explained which land use or cover land is linked to which service.

In the results and discussion, unexpected findings are not discussed, for example (page 12, paragraph 2, lines 3 and 4) “…. Unexpectedly, November rainfall alone showed a statistically significant increase at 99% confidence interval…”. There is no integrated discussion of the spatiotemporal dynamics of the four ecosystem services (food production, water supply, raw material, and climate regulation) in response to the four variables related to climate and productivity (precipitation, temperature, evapotranspiration, NDVI) and its variability (mean annual, main and second season, maximum, minimum). Mainly, the trends of the results are compared with other studies, the results from different sections are rarely integrated. The strengths and limitations of the study are not discussed. Some limitations are mentioned in the conclusions.

SPECIFIC COMMENTS BY SECTION

Keywords

It is suggested that the words in this section be different from the words in the title.

Introduction

Page 1 Paragraph 1 Lines 1,2, 7 and 8. The opening and closing phrases are very similar in the information they provide. It is suggested to improve the information and messages.

P1P1L4. Missing dot (.) after [4]

P1P2L3-7 When it mentions “In Ethiopia, climate change and its manifestations in rainfall variability,….. and human health conditions“ There are different climatic conditions and different services, it is a very general sentence “…. have negatively affected ecosystem services…”, no mechanisms or processes are mentioned that support the pattern described. Also, when they mention “crop production” what crops? Do all crops in Ethiopia respond the same to precipitation or temperature? Furthermore, “human health conditions” is not an ecosystem service.

P1P4L3 Missing period (.) after [16-19]

P2P4 It is not clearly justified why those four key services were selected (food production, water supply, raw material, and climate regulation services). No further context is given.

Materials and methods

P3P1L4 What does kebeles mean? Are they districts?

Figure 1. Scale bars are missing from maps B and C, as well as the compass rose. On map A, the 17 neighboring districts of the national park cannot be distinguished.

P4P2L1-2 Did the discussion with local community, agricultural extension agents, and park workers take place in workshops or interviews?

P4P2L2 What crops are considered in food production? What is raw material provision within the context of the MzNP?

P4P2L5 What LULC was associated with food production? What LULC was associated with raw material? Which LULC was associated with climate regulatory services? It is also important to detail why these LULCs are associated with services within the local context. How ecosystem service values were estimated from the LULC maps? Only one land use or cover type were used for service or several?

P4P2L13 What is the resolution of the satellite images used to do the supervised classification? Is the resolution higher or lower than the climate variability grid data?

P5P5L1-2 Add space between rainfall(mm) and series(mm)

P6P1L1-2 Add space between series(mm)

P9P1L4-5 Define if it is very strong or strong “…and very strong positive/negative correlation (≥±.8).strong positive/negative correlation (≥±.8) …”

P10P1L Since the study site has two distinct rainfall seasons and rainfall affects the NDVI values (season of greatest greenness and growth), what period of the year (months or days) did the Landsat images cover to obtain the average NDVI?

Results and discussion

In general, for this entire section it is suggested to be consistent in the order in which the results of the services are presented (food production, water supply, raw material, and climate regulation services), because it usually varies between tables and between figures. It is suggested to present them according to the order in which they are mentioned from the introduction.

P12P2L1-3 In Table 2, the data do not match the decreasing trend of April and June, as well as mean annual, and the main rainy season rainfall.

P12P3L3 in “Mk” the “K” is capitalized

P13P1L1-3 The information “…..According to [46], if …… 1:1 straight line….” It is presented in the materials and methods (P7P4L8-11). It is suggested to leave the explanation in only one of the two sections.

P15P3L3. The paragraph mentions the correlation between mean evapotranspiration and ecosystem services, but the evapotranspiration map is not shown in Figure 8, there is no description of the spatial variability of this variable, as is done for precipitation and temperature. Furthermore, correlation results are not shown in Table 6. What is the reason?

Figure 9 So that the NDVI maps are visually comparable and easy to distinguish spatial and temporal variations, it is suggested that the color legend consider the most negative value (-0.37) and the most positive value (0.74), or failing that, that the color legend ranges from -1 to 1. For example, the color orange covers values from -0.37 to 0, while blue covers values from 0.50 to 0.74. Finally, the range of NDVI value per year is described in detail in the text (P16P1L3-5).

P18P1L1-9 Considering that the value of ecosystem services was derived from LULC maps and that the NDVI (a measure of greenness) that is highly correlated with NPP, plant cover, and ecosystem productivity (P17P1L1-2), was it obvious the high correlation shown in the results of Table 6?.

P19P1L15-7 It is suggested to discuss the result “…On the other hand, the climate regulation service in the study area exhibited a negative correlation with the minimum and maximum temperature…” as is done with other services.

P19P3L1-2 It is suggested to present the information on raw material services from when it is first presented in the introduction “…includes the production of lumber, fuel, or fodder that are extracted as raw materials ..”

P20P2 The results of the evapotranspiration and services correlations are very far from where they were first mentioned (page 15).

Supporting information

S2 and S3 Figure. It is suggested to add the P value (significance level) to the functional relationships between years and rainfall and temperature, not just the R square value.

6. PLOS authors have the option to publish the peer review history of their article (what does this mean?). If published, this will include your full peer review and any attached files.

Reviewer #1: No

Reviewer #2: No

---

## [Author Response · Author response to Decision Letter 0]

10 Jul 2024

Responses for Revisions of Manuscript

Manuscript Number: PONE-D-24-05650

Title: Spatiotemporal dynamics of ecosystem services in response to climate variability in Maze national park and its environs, southwestern Ethiopia

Journal: PLOSONE

Dear editor and reviewers,

Thank you for your valuable time and effort in reviewing our manuscript and providing insightful comments and suggestions. We have carefully considered all your comments, and almost all of them have been accepted and incorporated into the revised manuscript. To strengthen the literature review, highlight the novelty of our work, and enrich the discussion, we have reviewed an additional 13 articles. Changes made in the manuscript are marked using Track changes, and the newly added references are highlighted in blue in the text and reference section. We have provided point-to-point responses to your comments and highlighted them in blue as follows:

Thank you for your review

Journal Requirements:

1. When submitting your revision, we need you to address these additional requirements. Please ensure that your manuscript meets PLOS ONE's style requirements, including those for file naming. The PLOS ONE style templates can be found at https://journals.plos.org/plosone/s/file?id=wjVg/PLOSOne_formatting_sample_main_body.pdf andhttps://journals.plos.org/plosone/s/file?id=ba62/PLOSOne_formatting_sample_title_authors_affiliations.pdf.

Response: Thank you for the suggestion. We have adjusted our manuscript according to the journal’s requirements.

Response: Thank you. We did not generate any new code for this study. However, we utilized pre-developed codes for the Mann-Kendall test, Sen's slope estimator, ITA tests, and for generating temperature and rainfall trend graphs in R statistical software.

Response: Thank you for your comment. We have now provided similar grant information in both the ‘Funding Information’ and ‘Financial Disclosure’ sections. 

4. We note that your Data Availability Statement is currently as follows: [All relevant data are within the manuscript and its Supporting Information files]. Please confirm at this time whether or not your submission contains all raw data required to replicate the results of your study. Authors must share the “minimal data set” for their submission. PLOS defines the minimal data set to consist of the data required to replicate all study findings reported in the article, as well as related metadata and methods (https://journals.plos.org/plosone/s/data-availability#loc-minimal-data-set-definition).

Authors do not need to submit their entire data set if only a portion of the data was used in the reported study. If your submission does not contain these data, please either upload them as Supporting Information files or deposit them to a stable, public repository and provide us with the relevant URLs, DOIs, or accession numbers. For a list of recommended repositories, please see https://journals.plos.org/plosone/s/recommended-repositories.

Response: Thank you. We have uploaded data used for rainfall and temperature variability analysis, trend tests (Tables and graphs), and points generated from images for spatial analysis in supporting information files.

5. Please upload a copy of S1 Figure, S2 Figure, S3 Figure and S1 Table to which you refer in your text on page 33. Please amend the file type to 'Supporting Information'. If the Supplementary file is no longer to be included as part of the submission please remove all reference to it within the text.

Response: Thank you for the comment. Copies of S1 Figure, S2 Figure, S3 Figure, S1 Table, and S2 Table have been uploaded in supporting information files.

Responses for Reviewers' comments:

Reviewer #1

Abstract:

 Comment 1: The abstract summarizes the study effectively, but lacks specific quantitative findings.

Response: Thank you very much for the comment. Due to the word limit, we have now included some major quantitative findings in the abstract section, lines 39 to 48.

Introduction:

The introduction provides a comprehensive background on the importance of ecosystem services and the impact of climate variability. It effectively highlights the relevance of the study by emphasizing the lack of empirical research on this topic in Ethiopia. The introduction outlines the specific objectives of the study, aiming to fill this research gap, by focusing on the spatiotemporal impacts of climate variability on key ecosystem services in MzNP. However, there are some critical points to be improved:

Comment 2: The references are appropriate but could be expanded to include more recent studies on similar topics globally to strengthen the argument for the study’s novelty.

Response: Thank you for this comment. In order to strengthen the literature review, we have included additional four (4) recent studies in the introduction section in page 1 paragraph 1, lines 60 to 72 and page 2 paragraph 2, lines 102 to 111.

Comment 3: The identification of research gap is well articulated, but it could benefit from more detailed discussion on why previous studies were insufficient.

Response: Thank you very much. We addressed the research gap in the second paragraph of page 2, lines 98 to 111, by comparing our study with previous research on ecosystem good and services. Previous studies primarily focused on the impacts of LULC changes on ecosystem services, overlooking the effects of climate variability and lacking empirical analysis of these variables.

Comment 4: The text should consistently distinguish between different types of ecosystem services to enhance clarity.

Response: Thank you for this comment. Different types of ecosystem services were discussed according to the Millennium Ecosystem Assessment (MEA) classification in page 1, paragraph 1, lines 60 to 64.

Comment 5: It is necessary to use consistent terms for climate-related phenomena and impacts. For example, “climate change” and “climate variability” are used interchangeably but have distinct meanings.

Response: Thank you for the comment. While climate change and variability are distinct terms, the impacts of climate change often manifest through climate variability. However, we have now focused more on climate variability and also provided their distinctions in introduction section page 1, paragraph1, lines 67 to 72. 

Comment 6: To give some examples of the mechanisms by which climate change affects ecosystem services, will provide a more comprehensive understanding of the linkages and processes involved.

Response: Thank you very much. We have discussed the mechanisms by which climate change and variability affect ecosystem services on page 1, paragraph 2, lines 78 to 85. It is stated as follows: “The changing climate, evidenced by climate variability, affects the provision of ecosystem services and the well-being of people who rely on these services[12] . Ethiopia, like many other African countries, is very susceptible to the negative impacts of climate change and variability due to its low adaptive capacity[13]. In Ethiopia, climate change, characterized by high temperature, reduced rainfall, and increased rainfall variability and its manifestations in extreme weather conditions have negatively affected ecosystem services such as food production, groundwater availability, and soil organic matter and soil quality[14].”

Comment 7: The aims are quite broad, and the authors don’t provide specific research questions that guide the research. Defined aims and question would provide a sharper focus for the study.

Response: Thank you for the comments as well. In the introduction section on page 2, paragraph 2, lines 115 to 120, we have now included the specific objectives addressed in this study.

Methods:

Comment 8: The methods section is detailed and describes the study area, however, discuss specific characteristics of MzNP that make it a relevant case study for examining the impacts of climate variability on ecosystem services, such as its biodiversity, ecosystem types, or known climate vulnerabilities.

Response: Thank you for this comment. We have included more detailed descriptions of the study area, the ecosystem services obtained from there, and the major factors contributing to the decline of these services on page 3, paragraph 2, lines 130 to 132, and page 4, paragraph 1, lines 151 to 156.

Comment 9: The methods for collecting climate data are clear, but the explanation for ecosystem services data is vague. I suggest providing a more detailed explanation of how ecosystem services data were measured or quantified. I suggest specifying the indicators used, the data sources, and the methods of data collection.

Response: Thank you for the suggestion. The valuation of ecosystem services was based on Costanza et al. (1997) valuation method considering the contribution of each LULC classes for multiple ecosystem services. We applied benefit transfer method to the valuation of ecosystem services using recently developed ecosystem services value coefficients of Ecosystem Services Valuation Database (ESVD) (de Groot et al., 2020). The detailed explanation about ecosystem service value estimation from LULC classes using benefit transfer method was explained in our previous study (Simeon and Wana, 2024, pages 4 and 5) from which ecosystem service values of the study area in 1985, 2005, and 2020 were taken and similar procedure was followed for estimation of 1995 and 2015 values (page 5, paragraph 1 and 2, lines 180 to196).

Comment 10: The use of the MK trend test, Sen’s slope estimator, and ITA are appropriate for analyzing climate trends. The use of Pearson correlation and partial correlation to evaluate relationships among variables is also well-justified. However, while the description of the study area is thorough, including more specific reasons for choosing MzNP would strengthen the rationale. Finally, there should be a discussion on the assumptions underlying the statistical methods and how they were addressed, this will provide robustness to the methodology and ensure the validity of the results.

Response: Thank you very much. We have discussed the assumptions underlying the statistical methods employed in this study and how they were addressed in the methods section. The assumptions for the Mann-Kendall (Mk) trend test and Sen’s slope estimator were considered due to their insensitivity to outliers and lack of distribution assumptions. However, they may contain some error if autocorrelation exists in the time series data. Thus, autocorrelation was tested by calculating the autocorrelation coefficient at lag-1. To address this issue, the modified Mann-Kendall method was used to correct serially autocorrelated data with a significant lag-1 autocorrelation coefficient using the variance correction method in R statistical software (Page 7 paragraph 1 lines 230 to 238). Response: The ITA method is valid regardless of the sample size, serial correlation structure of the time series, and non-normal probability distribution of the data. Therefore, no underlying assumptions were employed (page 8, paragraph 3, lines 262 to 266). For Pearson and partial correlation analysis, since the distribution of the data affects parametric tests, histograms and normality tests were employed to determine whether the distribution of the data was normal. Non-normally distributed data were then transformed using Log10 data transformation techniques (page 9 paragraph 2 lines 283 to 286).

Results:

Comment 11: The results are presented clearly, with the findings from the MK test, ITA and correlation analyses well-organized. The results show significant trends in climate variables and their correlations with ecosystem services. However, visual aids (e.g, graphs, charts) could enhance the clarity and impact of the findings. While the results are clear, the interpretation could be more nuanced. I suggest a time series graph showing temperature and rainfall trends over the study period would provide a clearer picture of the data.

Response: Thank you for the suggestion. Time series graphs of the mean annual, main rainfall season, and second rainfall season, as well as the mean annual, maximum, and minimum temperatures, have now been attached as supporting information files S2 and S3.

Discussion:

Comment 12: The discussion section interprets the results in the context of existing literature, emphasizing the implications for ecosystem services and climate adaptation strategies. It addresses the study’s limitations and suggests areas for future research. However, while the results are clear, the interpretation could be more nuanced. I suggest discussing potential reasons for the observed trends and correlations would add depth. This could provide a broader context for results. I suggest discussing potential ecological or social factors that might explain the observed trends and correlations. For example, consider the role of land-use changes, conservation policies, or socio-economic conditions in the region.

Response: Thank you for the comment. We have incorporated the potential reasons for the observed variabilities, trends and correlations. For instance, we explained the reasons for high rainfall variability (page 12, paragraph 2, lines 375 to 377), temperature variability (page 15, paragraph 1, lines 444 to 446), the correlation between minimum and maximum temperature and raw materials (page 21, paragraph 2, lines 595 to 601), and the correlation between minimum and maximum temperature and climate regulation services (page 21, paragraph 3, lines 608 to 614).

Comment 13: There is limited comparison with findings from other regions. This will help place the results in a broader context and highlight similarities or differences in climate impacts on ecosystem services across different regions.

Response: Thank you very much. Additional previous studies from different continents and regions are included for comparison with our findings. These are presented on page 12, paragraph 1, lines 369 to 371; page 13, paragraph 2, lines 405 to 408; page 16, paragraph 1, lines 455 to 457; page 21, paragraph 1, lines 598 to 601; and page 21, paragraph 2, lines 608 to 611.

Comment 14: In general, the discussion is insightful but could benefit from deeper analysis. I suggest exploring the mechanisms driving the observed trends and correlations in greater detail. For example, discuss how specific climatic factors, such as seasonal rainfall patterns or temperature extreme, affect different ecosystem services like water provisioning, biodiversity, or carbon sequestration.

Response: Thank you for the comment. We have discussed how rainfall, temperature, altitude, and NDVI distribution affect ecosystem services on page 19, paragraph 2, lines 539 to 542; page 21, paragraph 2, lines 593 to 598; and page 

---

## [Editor Report · Decision Letter 1]

16 Jul 2024

Spatiotemporal dynamics of ecosystem services in response to climate variability in Maze National Park and its environs, southwestern Ethiopia

PONE-D-24-05650R1

Dear Dr. Simeon,

We’re pleased to inform you that your manuscript has been judged scientifically suitable for publication and will be formally accepted for publication once it meets all outstanding technical requirements.

Kind regards,

Angelina Martínez-Yrízar, Ph.D.

Academic Editor

PLOS ONE

Additional Editor Comments (optional):

Additonal minor corrections:

To ensure consistency in Table 2, please correct the labels of the last two columns, which currently read LCL/UCL, while the note at the bottom states UCL/LCL.

In Table 4, please change the first letter of the caption to uppercase.

Table 8 is incorrectly placed; it is referenced on page 19 but is located on page 22.
---

## [Editor Report · Acceptance letter]

18 Jul 2024

PONE-D-24-05650R1 

PLOS ONE

Dear Dr. Simeon, 

I'm pleased to inform you that your manuscript has been deemed suitable for publication in PLOS ONE. Congratulations! Your manuscript is now being handed over to our production team.

Kind regards, 

on behalf of

Dr. Angelina Martínez-Yrízar 

Academic Editor

PLOS ONE